# Modelling the evolution of Djankuat Glacier, North Caucasus, from 1752 until 2100 AD

Yoni Verhaegen[1], Philippe Huybrechts[1], Oleg Rybak[1,2,3] and Victor V. Popovnin[4]

[1]Earth System Science and Department of Geography, Vrije Universiteit Brussel, Pleinlaan 2, B−1050 Brussels, Belgium
[2]Water Problems Institute, Russian Academy of Sciences, Gubkina Str. 3,119333 Moscow, Russia
[3]FRC SSC RAS, Theatralnaya Str., 8-a, 354000, Sochi, Russia
[4]Department of Geography, Lomonosov Moscow State University, 1 Leninskie Gory, 119991 Moscow, Russia

*Correspondence to*: Yoni Verhaegen (yoni.verhaegen@vub.be)

**Abstract**. We use a numerical flow line model to simulate the behaviour of the Djankuat Glacier, a WGMS reference glacier
situated in the North Caucasus (Republic of Kabardino-Balkaria, Russian Federation), in response to past, present and future
climate conditions (1752−2100 AD). The model consists of a coupled ice flow−mass balance model that also takes into
account the evolution of a supraglacial debris cover. After simulation of the past retreat by applying a dynamic calibration
procedure, the model was forced with data for the future period under different scenarios regarding temperature,
precipitation and debris input. The main results show that the glacier length and surface area have decreased by ca. 1.4 km
(ca. -29.5 %) and ca. 1.6 km² (-35.2 %) respectively between the initial state in 1752 AD and present-day conditions. Some
minor stabilization and/or readvancements of the glacier have occurred, but the general trend shows an almost continuous
retreat since the 1850s. Future projections using CMIP5 temperature and precipitation data exhibit a further decline of the
glacier. Under constant present-day climate conditions, its length and surface area will further shrink by ca. 30 % by 2100
AD. However, even under the most extreme RCP 8.5 scenario, the glacier will not have disappeared completely by the end
of the modelling period. The presence of an increasingly widespread supraglacial debris cover is shown to significantly delay
glacier retreat, depending on the interaction between the prevailing climatic conditions, the debris input location, the debris
mass flux magnitude and the time of release of debris sources from the surrounding topography.

## 1 Introduction

Recently, a lot of attention has been given to modelling mountain glaciers, in particular due to their worldwide observed
shrinkage and important role within the current changing climate (e.g. Shannon et al., 2019; Zekollari et al., 2019; Hock et
al., 2019). The observed warming trend is a significant matter of concern to scientists and all other people (in)directly
involved in the behaviour of these glacial systems, as projected scenarios point towards an even further increase of the global
mean temperature in the future, especially if no efficient mitigation strategies are implemented (Stocker et al., 2013; Rasul
and Molden, 2019; Hock et al., 2019). Being consistent with this global trend, the accelerated retreat of Caucasian glaciers
during the last several decades has been clearly noticed (e.g. Shahgedanova et al., 2014; Zemp et al., 2015; Tielidze, 2016).

Accordingly, total glaciated area has decreased from 691.5 ± 29.0 km² to 590.0 ± 25.8 km² (-0.52 % yr⁻¹) in the period between 1986 and 2014 (Tielidze et al., 2020). Further degradation of Caucasian glaciers may affect the supply of water used for drinking, irrigation and hydroelectric energy generation, whereas it may also pose a threat for downstream communities via flooding, glacier collapses, avalanches, debris flows and glacial lake outbursts (e.g. Volodicheva, 2002; Ahouissoussi et al., 2014; Taillant, 2015; Chernomorets et al., 2018). Furthermore, the presence of glaciers in the Caucasus can be considered important for paleoclimatic research, tourism, cultural heritage and biodiversity (e.g. Popovnin, 1999; Shahgedanova et al., 2005; Hagg et al., 2010; Makowska et al., 2016; Tielidze and Wheate, 2018; Rets et al., 2019). Despite these rising concerns, however, modelling of Caucasian glaciers is scarce and has only been attempted in a few studies (e.g. Rezepkin and Popovnin, 2018; Belozerov et al., 2020).

In a warming climate, debris coverage onto the glacier's surface is believed to increase drastically due to the build-up of more englacial melt-out material, lower flow velocities and increased slope instability, which favour the occurrence of rock slides and mass movements from the surrounding topography (Østrem, 1959; Kirkbride, 2000; Stokes et al., 2007; Jouvet et al., 2011; Carenzo et al., 2016). During the last decades, a sharp increase of debris-covered glacier surfaces has been observed over the Caucasus region, owing to the combined effects of steep terrain, a wet climate, small average glacier size, large lateral moraines and the presence of local easily erodible sedimentary rock outcrops. Accordingly, debris coverage has expanded at a rate of ca. +0.22 % yr⁻¹ between 1986 and 2014 when the entire Caucasus region is considered (Tielidze et al., 2020). Scherler et al. (2018) estimate the supraglacial debris cover on Caucasian glaciers to be 26.2 % (ca. 155 ± 6.7 km²) at present-day, hence enabling the area to hold the world's most abundant share of debris-covered glacier surfaces in relative terms. Evidently, the presence of such supraglacial debris can influence the evolution of mountain glaciers in a variety of ways, depending on its thickness, properties and spatial/temporal distribution (Nicholson and Benn, 2006; Anderson and Anderson, 2016). Apart from a slight melt enhancement for a very thin debris layer, thick debris has been shown to reduce runoff volumes and reverse mass balance gradients due to its melt-reducing effect (e.g. Østrem, 1959; Bozhinskiy et al., 1986; Anderson and Anderson, 2016). If a thick supraglacial debris cover is present over a large portion of a glacier's ablation zone, surface melting and terminal retreat can be drastically suppressed, even under a warming climate (e.g. Scherler et al., 2011; Benn et al., 2012). In such cases, debris-covered glaciers are shown to lose mass by lowering the surface in their ablation zone (downwasting), rather than by terminus retreat (e.g. Hambrey et al., 2008; Rowan et al., 2015). The pronounced effect of debris should therefore not be ignored in numerical experiments to determine the future evolution of mountain glaciers, yet only few studies have included this complex process in time-dependent models (e.g. Jouvet et al., 2011; Rowan et al., 2015; Huss and Fischer, 2016; Kienholz et al., 2017; Rezepkin and Popovnin, 2018; Wirbel et al., 2018).

In this paper, we focus on modelling the Djankuat Glacier (North Caucasus, Russian Federation), a WGMS (World Glacier Monitoring Service) reference glacier which has a broad observational network in both space and time. However, despite abundant field data availability and increasing interest concerning its future behaviour, the Djankuat Glacier has not yet been modelled extensively. Here, we present a numerical flow line model to simulate its response to past, present and future

climatic change. The calculations relate to ice dynamics, supraglacial debris cover evolution and annual surface mass balance. More specifically, the objectives of this study are to construct and calibrate a coupled ice flow−mass balance−supraglacial debris cover model for the Djankuat Glacier, to reconstruct its front variations and mass balance series since 1752 AD, and to simulate the response to future climate change under different scenarios until 2100 AD. In particular, we adapt a physically-based debris model from Anderson and Anderson (2016) to investigate the impact of supraglacial debris cover on the glacier's evolution, which has not been previously applied in time-dependent numerical flow line models. The results can hence be used to more accurately assess the behaviour of the Djankuat Glacier as a WGMS reference glacier for the Caucasus area, including the potential side effects of its evolution such as the regulation of water resources. Furthermore, the refined debris cover implementation can be used for comparable glacier models in future research.

## 2 Location, data and models

### 2.1 The Djankuat Glacier

The Djankuat Glacier [43°12' N, 42°46' E] is a northwest-facing and partly debris-covered temperate valley glacier that is situated on the northern slope of the Main Caucasus Ridge near the border of the Russian Federation with Georgia, which is the most heavily glaciated area of the Northern Caucasus Mountains. As of 2010 AD, the glacier consists of four major ice flows and had a length of 3.26 km when taken from its highest point on the south face of the Djantugan peak (Fig. 1). The glacier occupied a total surface area of 2.688 km², of which the majority is situated above 3200 m ASL (Fig. 2). However, by 2017 AD, satellite imagery revealed that the glacier area had further decreased to 2.418 km$^2$ (Rets et al., 2019), while the glacier length shortened to a value of 3.12 km. Furthermore, a unique characteristic of the glacier is the origin of its main ice flux on the divergent and vast Djantugan firn plateau south of the main ridge, of which the contributing area to the glacier changes regularly (Aleynikov et al., 2002a).

The Djankuat Glacier has been monitored thoroughly since glaciological measurements began in the 1960s, resulting in an abundant amount of field data, enabling this glacier as an ideal candidate for modelling studies (e.g. Popovnin, 1999; Aleynikov et al., 2002b; Popovnin and Naruse, 2005; Lavrentiev et al., 2014; WGMS, 2018; Rets et al., 2019). Consequently, the Djankuat Glacier has been selected by the WGMS as a reference glacier for the Caucasus region, hence defining its behaviour as representative for other glaciers across this area. As such, a comparison with glacier length variations in the Caucasus since the 19[th] century AD shows that the Djankuat Glacier genuinely reflects the general trend in the broader area, as can be seen in Fig. 3 (e.g. Kotlyakov et al., 1991; Solomina et al., 2016; WGMS, 2018).

### 2.2 Field data

The start of the standard monitoring program on the Djankuat Glacier dates back to the 1967/68 AD season and includes measurements concerning geometry, supraglacial debris cover and (local) annual surface mass balance. Additionally, ice velocity measurements were performed during the summer seasons of 1994−2001 based upon both direct (theodolite surveys

of stakes) and indirect (stereophotogrammetrical) measurements, of which the resulting maps are reported in Aleynikov et al. (1999) and Pastukhov (2011). Glacier-wide ice thickness maps have also been constructed by Lavrentiev et al. (2014), using ground-based radio-echo measurements. However, direct and reliable observations lack at the higher elevations (> 3600 m) and the Djantugan Plateau due to difficult accessibility. In these areas, ice thickness values have been derived indirectly using surface velocity and slope (Aleynikov et al., 2002b; Pastukhov, 2011). The current ice thickness has been found to go up to ca. 100 m in the central part of the main glacier body, and to more than 200 m at the Djantugan Plateau. Furthermore, the glacier's cumulative surface mass balance during the 1967/68−2016/17 period exhibited a strongly negative value of -14.33 m w.e., with a mean equilibrium line altitude (ELA) of 3213 m (WGMS, 2018). Moreover, the mass balance profile in the upper areas is significantly modified (at 3600 m by ca. -76 % of the value that the specific mass balance would have if it were extrapolated according to the mass balance gradient found below) by snow redistribution processes (Pastukhov, 2011).

Both glacier-averaged debris thickness (from 0.28 m in 1983 to 0.54 m in 2010) and total debris-covered area (from ca. 0.10 km² or 3.5 % in 1968 to ca. 0.34 km² or 12.7 % of the glacier in 2010 AD) have increased largely. However, at the debris-covered left side of the snout (when seen in the downstream direction), debris thickness increased exponentially over the years, resulting in mean values of 1 m at the glacier front in 2010, compared to 0.29 m in 1983 and 0.45 m in 1994 AD (Popovnin et al., 2015). Recent observations have shown the importance of the debris cover on the Djankuat Glacier, as the debris-covered left side of the front clearly retreated slower than the less affected right part. As of 2010 AD, the length difference between both sides was ca. 180 m (Fig. 1) but this has increased to ca. 250 m by 2017 AD (Rets et al., 2019).

The climate around the glacier can be inferred from nearby weather stations, such as Terskol (elevation 2141 m, approx. 20 km northwest of the glacier) and Mestia (approx. 16 km southwest from Djankuat Glacier, in Georgia, at 1441 m elevation), see Fig. 1 and Table 2. The average mean annual temperatures in Terskol and Mestia are 2.6 °C and 6.0 °C respectively for the 1981−2010 reference climate. For the summer half-year from April to September (AMJJAS), the corresponding mean temperatures are 8.7 °C and 12.0 °C. Precipitation, on the other hand, is rather complex in the region due to variations of atmospheric circulation patterns, orographic uplift and convective precipitation in the summer season (Boyarsky, 1978; Shahgedanova et al., 2007; Hagg et al., 2010; Popovnin and Pylayeva, 2015). At Terskol and Mestia, total annual precipitation amounts equal 1001.1 and 1035.1 mm yr⁻¹ w.e. respectively for the 1981−2010 climate. During the accumulation season (October to March, ONDJFM), the corresponding precipitation values are 418.4 and 490.0 mm yr⁻¹ w.e., respectively. In 2007, two automatic weather stations (AWS) were additionally installed, one in the Adylsu Valley at ca. 2640 m elevation (AWS 1 in Fig. 1) and one in the ablation zone of the glacier at ca. 2960 m on a sparsely debris-covered ice surface (AWS 2 in Fig. 1). During the summer seasons (June to September, JJAS) of 2007−2017, a wide range of additional meteorological variables have therefore been acquired by both AWSs (air temperature, dew point temperature, incoming and outgoing shortwave/longwave radiation, relative humidity, wind speed and direction, air pressure and for AWS 1 also precipitation amounts). The AWSs did, however, not operate outside the JJAS period (Rets et al., 2019).

## 2.3 Ice dynamic model

The ice dynamic model is implemented as a numerical flow line model, in which the prognostic continuity equation for ice thickness change is solved. We choose to only model ice flow along a central axis in the x-direction and not upgrade the model to 3D due to the abundant amount of experiments that were conducted. However, the y-dimension is implicitly taken into account due to inclusion of glacier width along this central axis. One central flow line is considered with a total length of 5 km, stretching from the glacier top near the Djantugan peak down to the current snout and further into the Adylsu

Valley (Fig. 1). The flowline is constructed perpendicular to the surface elevation isolines, generally close to the location where the cross-sectional ice thickness and ice velocity are maximal, as determined from ice thickness and surface velocity maps. The model treats ice flow as a non-linear diffusion problem in a vertically integrated approach (e.g. Oerlemans, 2001):

$$\frac{\partial H}{\partial t} = -\frac{1}{W_{sfc}}\left(\frac{\partial F_{ice}}{\partial x}\right) + b_a$$

$$= -\frac{1}{W_0 + \mu H}\left[\frac{\partial}{\partial x}\left(\left[(W_0 + \frac{1}{2}\mu H)(\rho_i g H)^3 (f_d H^2 + f_s)\left(\frac{\partial h}{\partial x}\right)^2\right]\frac{\partial h}{\partial x}\right)\right] + b_a \qquad (1)$$

where $H$ is the ice thickness, $t$ the time, $\mu$ the slope of the lateral valley walls, $W_0$ the glacier bed width, $W_{sfc}$ the glacier surface width, $F_{ice}$ the ice volume flux, $x$ the horizontal distance, $b_a$ the local annual surface mass balance, $\rho_i$ the ice density, $g$ the gravitational acceleration, $f_d$ the flow parameter related to internal deformation, $f_s$ the flow parameter related to basal sliding and $h$ the surface elevation. The vertically integrated velocity is calculated by assuming that the 1D Shallow Ice Approximation is applicable to derive driving stresses on a xz plane and that ice is treated as a homogenous,

incompressible and isothermal non-Newtonian fluid in Glen's flow law. For basal sliding, a simplified Weertman-type flow law is used where the basal water pressure is proportional to the ice thickness and the basal shear stress equals the driving stress (e.g. Oerlemans, 1992; Oerlemans, 2001; Leclercq et al. 2012):

$$\overline{u} = \overline{u}_d + u_s = \left(-\rho_i g H \frac{\partial h}{\partial x}\right)^3 \left(f_d H + \frac{f_s}{H}\right) \qquad (2)$$

Here, $\overline{u}$ is the vertically averaged horizontal velocity, and $\overline{u}_d$ and $u_s$ are the velocity components related to internal

deformation and basal sliding respectively. Equation (1) is then solved on a staggered grid with a spatial resolution $\Delta x$ of 10 m. Integration over time is achieved with a forward in time, centered in space (FTCS) numerical scheme using a time step $\Delta t$ of 0.0005 years, as determined by the CFL-condition for diffusion problems.

## 2.4 Mass balance model

The mass balance model is based upon the difference between accumulation $ACC$ and runoff $RO$ over the balance year (1

October – 30 September), so that the local surface mass balance $b_a$ is defined as:

$$b_a = \int_{yr}(ACC - RO) * dt \qquad (3)$$

Mean specific (total) mass balances $B_a$ were then derived by integrating $b_a$ over the entire glacier surface. Accumulation for each point along the flow line is only dependent on the part of the total precipitation that is solid ($P_{solid}$), which only takes place if precipitation occurs below a certain threshold temperature $T_{tresh}$:

$$160 \quad ACC = P_{solid} = \begin{cases} ([P_{Terskol} * f_e] * P_{scale}) * f_{red} & if \ T_{air} < T_{tresh} \\ 0 & if \ T_{air} \geq T_{tresh} \end{cases} \tag{4}$$

Air temperatures $T_{air}$ from Terskol weather station were interpolated to any height on the Djankuat Glacier by applying vertical temperature lapse rates $\gamma_T$ (Table 1). A direct comparison of measured air temperatures between AWS 2 on Djankuat and the Terskol weather station was found to exhibit a strong correlation ($R^2 = 0.81$), generating a summer season lapse rate of -0.0067 °C m$^{-1}$ between 2007−2017 AD. Due to lack of AWS data outside of the JJAS period, a temperature lapse rate of -0.0049°C m$^{-1}$ was used for the winter half-year (ONDJFM), in accordance with a mean annual ELA temperature of -3.75 °C for Djankuat Glacier (WGMS, 2018). The term $P_{Terskol} * f_e$ represents the precipitation in the Adylsu Valley, calculated by multiplying the precipitation in Terskol with a horizontal precipitation enhancement factor $f_e$ to account for horizontal precipitation variations. In this study, a value for $f_e$ of 1.5 between Terskol and the Adylsu Valley was found after a comparison of precipitation amounts from AWS 1 in the glacier valley. The factor $P_{scale}$ is then used to scale the obtained precipitation amounts to the entire glacier from the Adylsu Valley to any surface elevation $h$, by making use of a vertical precipitation gradient $\gamma_P$, where the latter is used as a tuning parameter due to a lack of data (see Sect. 3.1):

$$P_{scale} = \left( \frac{P_{Terskol} * f_e + (\gamma_P * \Delta h)}{P_{Terskol} * f_e} \right) \tag{5}$$

At last, the factor $f_{red}$ represents a snow redistribution factor which corrects the solid precipitation for redistribution by wind and/or avalanches. Here, a topographic characteristic is used to parameterize snow addition or removal from the glacier surface. It was quantified by dividing the linear accumulation profile (without the redistribution factor) with the observed profile and correlating these anomalies to the laterally averaged surface slope $s$ along the flow line (e.g. Huss et al., 2009). As such, a polynomial fit was found. For slopes steeper than a threshold $s_{crit}$, removal of snow can occur, and is assumed to be influenced by the surface slope itself:

$$f_{red} = \begin{cases} 1.2 & if \ s < s_{crit} \\ -0.0017s^2 + 0.0535s + 0.9041 & if \ s \geq s_{crit} \end{cases} \tag{6}$$

The critical slope $s_{crit}$ distinguishes between slopes $s$ that either favour snow addition or snow removal (Table 1). We do acknowledge that the $f_{red}$ parameterization is solely used for curve fitting of the accumulation profile.

Melt production $M$, on the other hand, only takes place when the net energy flux per unit area at the surface $\Psi_0$ is positive (e.g. Oerlemans, 2001; Nemec et al., 2009):

$$M = max \left( 0, \frac{\Psi_0}{\rho_w L_m} \right) \tag{7}$$

where $\rho_w$ is the water density and $L_m$ the latent heat of fusion. As discussed further in section 2.5, the melt term $M$ is further modified by the debris cover and meltwater retention in the snowpack to obtain the total runoff $RO$. The net energy flux is parameterized as (Oerlemans, 2001; Giesen and Oerlemans, 2010; Leclercq et al., 2012):

$$\Psi_0 = \begin{cases} S_\downarrow(1-\alpha)\tau + c_0 & if \ T_{air} < T_{break} \\ S_\downarrow(1-\alpha)\tau + c_0 + c_1 T_{air} & if \ T_{air} \geq T_{break} \end{cases} \tag{8}$$

Here, $\tau$ is the atmospheric transmissivity, $\alpha$ is the surface albedo, while $c_0$ and $c_1$ are constants to describe the air temperature-dependent fluxes (i.e. the net longwave, latent heat and sensible heat fluxes). Hence, for air temperatures below the threshold $T_{break}$, $\Psi_0$ has a constant value. For higher temperatures, however, $\Psi_0$ increases linearly with $T_{air}$, where the rate of increase is determined by $c_1$ (Giesen and Oerlemans, 2012). The downward incoming solar radiation at the surface $S_\downarrow$ incident on an inclined surface with a certain surface slope and aspect, is calculated as (e.g. Oerlemans, 2001):

$$S_\downarrow = \begin{cases} S_{\downarrow(TOA)}(f_{dir}\cos(\theta) + f_{dif}\cos(\theta_z)) & if \ \theta_e > 0° \ \& \ \theta < 90° \\ S_{\downarrow(TOA)}(f_{dif}\cos(\theta_z)) & if \ \theta_e > 0° \ \& \ \theta \geq 90° \\ 0 & if \ \theta_e \leq 0° \end{cases} \tag{9}$$

where $S_{\downarrow(TOA)}$ is the incoming instantaneous extraterrestrial shortwave radiation on a horizontal plane at the top of the atmosphere (TOA), $\theta_e$ and $\theta_z$ are the solar elevation and zenith angle respectively calculated using basic astronomical formulas (e.g. Iqbal, 1983; Allen et al., 2006; Duffie and Beckman, 2006), and $\theta$ is the angle of incidence, taking into account the surface geometry. Further, $f_{dir}$ and $f_{dif}$ are the fraction of direct and diffuse solar radiation, which are derived from parameterizations used by Oerlemans (1992, 2001, 2010) and Voloshina (2002) that use the fractional cloud cover $f_{cl}$:

$$\begin{cases} f_{dir} = 0.1 + 0.80 * (1 - f_{cl}) \\ f_{dif} = 0.9 - 0.80 * (1 - f_{cl}) \end{cases} \tag{10}$$

At last, surface albedo $\alpha$ is parameterized as (e.g. Oerlemans and Knap, 1998; Nemec et al., 2009):

$$\alpha = \alpha_{snow} + (\alpha_{ice} - \alpha_{snow}) \exp\left(\frac{-d_{snow}}{d^*_{snow}}\right) \tag{11}$$

where $\alpha_{snow}$ is the snow albedo, $\alpha_{ice}$ the ice albedo and $d^*_{snow}$ a characteristic snow depth.

Measurements of the incoming solar radiation from the AWS 2 were used to derive atmospheric transmissivity. These data were therefore compared to the theoretical maximum incoming solar radiation at the top of the atmosphere, calculated with standard astronomical formulas (e.g. Iqbal, 1983; Allen et al., 2006; Duffie and Beckman, 2006). Consequently, the overall atmospheric transmissivity $\tau$ in the summer season over the Djankuat Glacier could be deduced as an average of 0.53 (Table 1). The ice albedo $\alpha_{ice}$ can, according to raw data from the AWS 2, vary between 0.15 and 0.40 depending on the presence of water, moraine cover and other impurities and has an average value of 0.22, corresponding to moderately debris-loaded ice. Sparse snow-covered conditions during the ablation season causes $\alpha_{snow}$ to increase to the 0.40−0.90 range (mean 0.79). Next, values for $f_{dir}$ and $f_{dif}$ are derived from the parameterization of the fractional cloud cover $f_{cl}$ over the Djankuat

Glacier, using an approximately linear relationship between the cloud cover and the net longwave radiation balance (Voloshina, 2002), of which the latter was derived from measurements by AWS 2 on the glacier surface. The analysis points out that direct and diffuse solar radiation are approximately equally important for the glacier (Table 1). The constants $c_0$, $c_1$ and $T_{break}$, describing the air-temperature dependent fluxes and their relationship with the air temperature $T_{air}$ itself, are at

last derived from measurements of AWS 2 of the net longwave radiation, as well as from a parameterization of the sensible and latent heat fluxes via Kuzmin's method (Kuzmin, 1961; Toropov et al., 2017). Here, these fluxes are added up and analyzed against air temperature following the method of Giesen and Oerlemans (2010) and Leclercq et al. (2012), as can be seen from Eq. (8).

**2.5 Debris cover model**

The supraglacial debris cover on the Djankuat Glacier was parameterized in order to account for the effects of melt reduction under debris-covered ice. The debris thickness was approached with a steady deposit model adopted from Anderson and Anderson (2016), where debris input onto the glacier is generated from a fixed point on the flow line. In the model, debris thickness then changes according to either melt-out from debris-loaded ice (first term), the downstream advection of supraglacial debris (second term) and the input or removal of supraglacial debris on the glacier surface (third term):

$$\frac{\partial H_{debris}}{\partial t} = -\left(\frac{C_{debris}(\min(0,b_a))}{(1-\phi_{debris})\rho_{debris}}\right) - \left(\frac{\partial(u_{sfc}H_{debris})}{\partial x}\right) + I_{debris} \qquad (12)$$

Here, $H_{debris}$ is the debris thickness, $t$ the time, $C_{debris}$ the englacial debris concentration, $\phi_{debris}$ the debris cover porosity, $\rho_{debris}$ the debris rock density, $b_a$ the specific surface mass balance, $u_{sfc}$ the glacier surface velocity and $I_{debris}$ the input or removal of debris from the glacier surface. The advection equation (Eq. 12) is solved using a first order upwind scheme with $\Delta t = 0.01$ years, in accordance with the CFL-condition for advection problems. In the model, the factors $\phi_{debris}$ and $\rho_{debris}$

are constants in space and time and taken at 0.43 and 2600 kg m$^{-3}$ respectively (Bozhinskiy et al., 1986). For $C_{debris}$, we use a value of 1.05 kg m$^{-3}$, referring to a bulk debris concentration inside the ice of 0.12 % as found by the same authors for the Djankuat Glacier in the 1980s (Table 1). Also here, a constant value in space and time is assumed. Incorporating englacial debris pathways or the spatial distribution of englacial debris concentration would add more detail than warranted by the lack of reliable data regarding this value.

Next, at the debris input location $x_{debris}$, a steady debris flux per unit area $F_{debris}^{input}$ transmits material from the surrounding topography to the glacier by means of a debris deposition rate (m yr$^{-1}$), starting from $t_{debris}$ onwards. Here, $t_{debris}$ is defined as the time at which the topographic debris source firstly starts to release its mass flux towards the glacier surface. We set the debris input location $x_{debris}$ at 1680 m from the highest point (just below the ELA, at 88% of the distance between the terminus $x_L$ and the ELA $x_{ELA}$), since it is the furthest point up-glacier for which observed debris thickness values are

reported in Popovnin et al. (2015). It was chosen to keep the debris input location at a fixed position due to the general absence of direct observations regarding past (static or moving) topographic debris sources. However, a comparison of

present-day satellite imagery with those from the 1970s (Pasthukov, 2011) points out that the debris patches exhibited only minor up-glacier migration on the main glacier tributary and the debris-covered part of the snout, lending some support to this assumption.

To avoid the buildup of unrealistically high debris thickness in low flow velocity zones in the future, we furthermore choose to let the debris mass flux stop when the surface width at point $x_{debris}$ has reached a value lower than 90 % ($t_{Wsfc-10\%}$) of its original value at time $t_{debris}$. This is considered a reasonable value, as the current observed debris-covered area is ca. 10 % at this specific point (Fig. 2). Connectivity issues between the topographic source and the main glacier are forwarded as the main reason to justify this modification of the Anderson and Anderson (2016) model. Consequently, by then the glacier

has laterally shrunk too much to ensure that debris fluxes could still reach its surface. At the terminus (the last non-zero ice thickness grid point), debris is removed into the foreland by a debris flux per unit area $F_{debris}^{x=L}$ (Anderson and Anderson, 2016):

$$I_{debris} = \begin{cases} F_{debris}^{input} & if\ x = x_{debris}\ \&\ t_{debris} \leq t < t_{Wsfc-10\%} \\ -F_{debris}^{x=L} = c_T H_{debris}^{x=L} & if\ x = x_L \\ F_{debris}^{x=L+1} = F_{debris}^{x=L(orig)} - F_{debris}^{x=L} & if\ x = x_{L+1} \\ 0 & else \end{cases} \tag{13}$$

where $x_L$ is the terminus position and $c_L$ is a constant describing the strength of debris removal from the terminus into the

foreland, for which we used the same value as suggested in Anderson and Anderson (2016), i.e. $c_L = 1$ (Table 1). As such, what is deposited in the foreland by $F_{debris}^{x=L+1}$ is the difference between the original debris flux on point $x = x_L$ (i.e. without the parameterization) minus the actual debris flux obtained with the parameterization. Eventually, the debris-related melt reduction factor $f_{debris}$ is taken as (e.g. Vacco et al., 2010; Huss and Fischer, 2016):

$$f_{debris} = \exp\left(\frac{-H_{debris}}{H_{debris}^*}\right) \tag{14}$$

Here, $H_{debris}^*$ is a characteristic debris thickness (i.e. the debris thickness at which the melt rate is $e^{-1}$ or ~ 37% of the clean ice melt rate). It must be noted that the melt enhancement that may occur for a very thin debris cover was not implemented in the debris model. However, values in literature of the debris thickness for which a maximum amount of melt enhancement occurs on the Djankuat Glacier vary from 0.02 m to 0.07 m (Bozhinskiy et al., 1986; Popovnin and Rozova, 2002; Lambrecht et al., 2011), and the areal fraction of Djankuat Glacier that holds these thin thickness values is very small

(Popovnin and Rozova, 2002; Popovnin et al., 2015). It is therefore not believed to have a significant influence on the ablation of Djankuat Glacier.

Next, the fractional debris covered area along the flow line is parameterized based upon the distance from the terminus $D_T$, for which an exponential relationship was found from observations that can, of course, not exceed 1:

$$\frac{A_{debris}}{A} = \min(G_A \exp(-0.01612 * D_T - 0.01720), 1) \tag{15}$$

Here, $G_A$ is a yearly updated growth factor that controls the expansion of the debris covered area (see Eq. 17 in Sect. 3.2). It is furthermore worth noting that the debris model also neglects other processes that may potentially play a role in the spatial and temporal distribution of debris, such as the formation and thickening of medial moraines, ice cliffs and surface ponds (Anderson and Anderson, 2016).

## 2.6 Calculation of runoff

In the case that snow is present at the glacier surface, runoff is calculated as the meltwater outflow from a saturated snowpack $W_{snow}$, following the principles applied in Schaefli and Huss (2011). On the other hand, in case of snow-free conditions, runoff is affected by the presence of a debris cover on the glacier ice (e.g. Lambrecht et al., 2011):

$$RO = \begin{cases} W_{snow} = max(0, w_{snow} - \eta_s d_{snow}) & if \ d_{snow} > 0 \\ M_{ice} = M\left(\frac{A - A_{debris}}{A}\right) + M\left(\frac{A_{debris}}{A}\right) f_{debris} & if \ d_{snow} = 0 \end{cases} \tag{16}$$

where $M$ is the melt production (see Sect. 2.4), $W_{snow}$ is the water outflow from the saturated snowpack, $w_{snow}$ the liquid
snow store, $\eta_s$ the water holding capacity of the snowpack, $f_{debris}$ the melt-reduction factor from debris, $A_{debris}$ the debris covered area and $d_{snow}$ the snow depth.

## 3 Model setup and calibration

### 3.1 Mass balance model

We used the 1967/68−2006/07 period to calibrate the mass balance model, as this time frame holds both specific (elevation-
dependent) and mean specific (glacier-wide) surface mass balance measurements (WGMS, 2018). Accordingly, 3-hourly temperature and precipitation data of the corresponding period were used from the Terskol weather station. For geometric data that serve as input for solar geometry calculations, we use laterally averaged values for slope and aspect, calculated by averaging all intra-glacier values along a line perpendicular to the flow line. Surface elevations were directly extracted from a DEM for 2009/10 AD conditions. We hereby take into account the same spatial spacing of 10 m that is used in the flow
model. Afterwards, geometric input data were smoothed using a window size of ± 100 m around every grid point. Calibration of the mass balance model further assumes the geometry (slope, aspect, glacier length and surface area) to be fixed over the 1967/68−2006/07 period, whereas in fact length and surface area decreased by 113 m and 0.346 km² respectively.

For the accumulation part, the vertical precipitation gradient $\gamma_P$ was used as a tuning parameter by fitting the accumulation
profile of the glacier. In literature, several values for this parameter have been proposed, varying between 0.0005 and 0.0046 m yr⁻¹ w.e. m⁻¹ (e.g. Boyarsky, 1978; Hagg et al., 2010; Giesen and Oerlemans, 2012; WGMS, 2018). To ensure successful calibration, a precipitation gradient of 0.0023 m yr⁻¹ w.e. m⁻¹ was derived to extract these data over the entire glacier surface. At last, the snow redistribution factor $f_{red}$ was used for curve fitting of the accumulation profile, as discussed before.

Concerning ablation, three variables were chosen as tuning parameters. Due to lack of field data concerning the water holding capacity of snow $\eta_s$, it was used to calibrate the ablation in the accumulation area. Additionally, the intercept of the air temperature-dependent fluxes $c_0$ was chosen as a second tuning parameter, again due to the lack of reliable and/or sufficient data during the observational period (Table 1). Next, for the factor $H^*_{debris}$ which controls the strength of the melt-reducing effect of debris, several values have already been proposed for the Djankuat Glacier (e.g. Bozhinskiy et al., 1986; Popovnin and Rozova, 2002; Lambrecht et al., 2011). Due to the large uncertainty, it was used as a third tuning parameter, this time for the lower elevation areas. Here, a value of 1.15 m was found to exhibit the best fit with the observations. We acknowledge that this value implies that the gradient of the exponential decay in Eq. 14 is somewhat out of range with respect to earlier studies for other glaciers (e.g. Anderson and Anderson, 2016). This rather atypical value can however be linked to the relatively high thermal conductivity of the granite-type debris cover on the glacier (2.8 W m⁻¹ °C⁻¹) and the high debris cover porosity (0.43 for Djankuat Glacier, Bozhinskiy et al., 1986). Also, the relatively low water saturation and large particle size, as suggested by Lambrecht et al. (2011), may imply that heat conduction towards the debris-ice interface seems to occur efficiently on the Djankuat Glacier.

With the calibrated surface energy balance model, the multiyear mean mass balance profile of the Djankuat Glacier during the 1967/68–2006/07 period is successfully reproduced, as the calculated mass balance vs. elevation profile matches nicely with the observations (Fig. 4). This profile reflects the determining processes affecting the Djankuat Glacier's mass balance: in the higher elevations, snow redistribution by wind/avalanches and meltwater retention are important factors, whereas in the lower areas, the presence of a supraglacial debris cover reduces the glacier's runoff volume significantly and hence dampens the mass balance gradient. Modelled mean specific balances of the Djankuat Glacier show a moderate agreement with observed values since 1967/68 AD ($R^2 = 0.52$). The RMSE of the multiyear mean mass balance-elevation profile and the individual local annual mass balances was reduced to 0.18 m yr⁻¹ w.e. m⁻¹ ($R^2 = 0.99$) and 0.61 m yr⁻¹ w.e. m⁻¹ ($R^2 = 0.91$) respectively (Fig. 4a and b).

As a remark, it must be noted that the calibration dataset for the mass balance model is quite long (39 years from 1967/68 to 2006/07 AD), making it credible to assume that the parameters calibrated to this period have some validity for past and future conditions as well. Apart from the high-elevation areas, where data availability is limited and snow redistribution processes create complex conditions (> 3600 m, of which the areal fraction is only ca. 3 % of the glacier area in 2010 AD), it can be expected that the environmental setting within the calibration window also holds for periods prior to and after the observational period. It must furthermore be noted that there are only few independent data to validate our model results with a sufficient degree of certainty.

### 3.2 Debris cover model

For the debris model calibration, we matched the temporal evolution of the average debris thickness at the front (i.e. the first 30 grid points) as well as the debris covered area, using $t_{debris}$, $F^{input}_{debris}$ and $G_A$ as tuning parameters. Values for the observed

debris cover at different elevation bands from the survey year 1968 AD (only for debris area) as well as for 1983, 1994 and 2010 AD (for both debris area and thickness) are available from Popovnin et al. (2015). Moreover, to obtain more detailed information concerning the current debris covered area on a spatial scale, the debris cover extent was manually digitized based on satellite imagery of the year 2010 (see Fig. 1).

Accordingly, the observed debris thickness evolution was found to be best reproduced by setting $t_{debris}$ to 1958 and $F_{debris}^{input}$ to 1.60 m yr$^{-1}$ (Table 1). At last, a power relation (R² = 0.85) was found between the growth factor $G_A$ and the modelled mean debris thickness at the glacier front as obtained in the previous step:

$$G_A = 1.17048 * \left(H_{debris}^{front}\right)^{0.62047} \tag{17}$$

Where $H_{debris}^{front}$ is the modelled debris thickness at the front (i.e. the first 30 grid points) as obtained before. As such, the
RMSE between modelled and observed values between 1967/68 and 2009/10 AD was reduced to 0.07 m (R² = 0.83) for debris thickness at the front and 0.9 % (R² = 0.95) for the fractional debris-covered area respectively (Fig. 5).

### 3.3 Ice dynamics model

To calibrate the flow model, it was initially run from zero ice thickness until a steady state situation was reached, which is achieved when the glacier has less than 0.002 % change in its total volume per year. The parameters $f_d$ and $f_s$ were adopted
to minimize the RMSE between observed and modelled ice thickness for 2010 AD conditions, assuming a steady state. Geometric input data for the flow model were extracted from a DEM for 2010 AD conditions. Hence, bedrock elevation was derived in combination with ice thickness maps from Pastukhov (2011) and Lavrentiev et al. (2014). Surface width was extracted by measuring the intra-glacier distance of 10-m spaced lines perpendicular to the orientation of the flow line. After extracting the lateral valley slopes, the width at the bed was calculated assuming a trapezoidal valley shape (e.g. Oerlemans,
1992; Gantayat et al., 2017). All data were finally joined to the closest point on the flow line for every 10 m and smoothed with a window of ± 100 m around every grid point. For the Djankuat Glacier, the best fit was found for $f_d = 6.5$ x $10^{-17}$ Pa$^{-3}$ yr$^{-1}$ and $f_s = 3.25$ x $10^{-13}$ Pa$^{-3}$ m² yr$^{-1}$ (Table 1). Additionally, the bed width for the assumed trapezoidal-shaped cross section was slightly adjusted to ensure that the parameterization fits the observed area-elevation distribution for a total surface area of 2.688 km². The full set of parameter values used in the model is given in Table 1. The steady state situation of
the ice flow model was at last tested by comparing the ice flux with the integrated upstream mass balance, by ensuring that the integrated surface mass balance over the entire glacier approaches 0 to within an acceptable accuracy (0.006 m yr$^{-1}$ w.e.), and by calculating the volume change with time. As expected for the model setup, all results exhibited an appropriate steady state situation for the glacier.

The flow model for the Djankuat Glacier was able to produce a steady state glacier profile with a length of 3.26 km after 200
years (Fig. 6a). The model approaches the observed ice thickness as it minimizes the RMSE to 14.27 m (R² = 0.90), see Fig. 6c. Despite minimized RMSE, the mismatch near the snout and steep slopes of the Djantugan peak increase the error of the

model. However, it is argued that a significant part of the error reflects either the current non-steady state situation of the glacier, the presence of a supraglacial debris cover at the front, or the lack of reliable and direct ice thickness observations at the highest elevations of the glacier. As with the mass balance and debris cover model, there are no, or only few, independent data to validate our model results with a sufficient degree of certainty.

Modelled current surface velocity for the Djankuat Glacier goes up to ca. 80 m yr$^{-1}$ near the ice falls of the Djantugan Plateau and also peak in the middle section of the glacier (Fig. 6d), which fits well with observations of maximum velocities in the 60−80 m yr$^{-1}$ range (Aleynikov et al., 1999; Pastukhov, 2011). Moreover, the modelled deformational and basal sliding components comprise respectively 45 % and 55 % of the vertically averaged ice flow velocity along the flow line.

## 4 Basic sensitivity experiments

With the calibrated submodels, some basic sensitivity tests were conducted which all initially started from a steady state glacier resembling the present-day geometry. Perturbed mass balance profiles (in steps of 0.25 m yr$^{-1}$ w.e.) were subsequently used as forcing into the flow model, until a new steady state was reached. As such, a relationship with a slight deviation from linear was found between the steady state length and the mass balance perturbations $\Delta B_a$, exhibiting a value of ca. 1100 and 1355 m (m yr$^{-1}$ w.e.)$^{-1}$ for negative and positive perturbations respectively (Fig. 7a). On the other hand, the e-folding length response time (i.e. the time needed to achieve 1 - e$^{-1}$ or ~63% of the total length change) of Djankuat is in the order of 31 ± 3 years. Additional sensitivity experiments with the mass balance model show that the Djankuat Glacier, when its 2010 AD geometry and other parameters are considered fixed, is quite sensitive to both temperature (-0.70 m yr$^{-1}$ w.e. °C$^{-1}$) and precipitation changes (0.20 m yr$^{-1}$ w.e. 10 %$^{-1}$). As such, a 1 °C annual temperature change for the Djankuat Glacier is only compensated when the precipitation change is in the order of ca. 35 %. Mass balance sensitivity to temperature changes shows a non-linear behavior, whereas the relationship is linear for precipitation changes (Fig. 7b).

To assess the climate and glacier sensitivity for equilibrium conditions, mass balance profiles were furthermore altered by temperature and precipitation perturbations within the -3 to +3 °C and -25 % to +25 % range respectively (as compared to the 1967/68−2006/07 AD reference values). Sensitivity of steady state length to temperature changes was found to exhibit a linear behaviour (815 m °C$^{-1}$) for perturbations between -1.4 and +0.7 °C, but is modelled to vary between 400 and 1400 m °C$^{-1}$ when assessed over the entire range (Fig. 7c). The glacier sensitivity depends largely upon geometry and increases (decreases) for more negative (positive) mass balance perturbations, predominantly due to the flatter (steeper) terrain. The sensitivity also peaks around a temperature perturbation of +1 °C, i.e. when the glacier front is positioned at the transition between the broad accumulation area and the narrower snout (ca. x = 2300 m on the flow line). Also, the non-linear nature of the temperature-mass balance relationship (Fig. 7b) triggers a deviation from linear behaviour. Consequently, the change in forcing needed for a retreat from 2 to 1 km is nearly twice as large as for a retreat from 4 to 3 km. For precipitation the sensitivity is more or less constant for a value of 250 m 10 %$^{-1}$ (Fig. 7d). A temperature increase of +3.4 °C compared to the

1967/68−2006/07 AD Terskol mean of +2.5 °C is sufficient to cause a total drawdown of the glacier, as the last ice on the Djantugan Plateau melts away 470 years after the induced perturbation.

## 5 Past reconstruction of the Djankuat Glacier

### 5.1 LIA extent of the glacier

All three submodels (ice flow, mass balance and debris cover) are finally coupled to determine the past and future evolution of the Djankuat Glacier. Here, the mass balance model and debris cover model calculate annual surface mass balance profiles, which are then used as input for the continuity equation in the ice flow model after conversion to ice equivalents. Glacier length $L$ is calculated by multiplying the number of non-zero ice thickness grid points by $\Delta x$. $L$ is thus not necessarily equal to the glacier terminus position, as the glacier may disintegrate in several sections during retreat. As a first step, the model is initialized with a spin up run in which a steady state glacier, as well as a steady state debris cover, are produced for the balance year 1752/53 AD. Although we have no clear indication to suspect steady state behaviour at this time due to lack of reliable data on debris cover, mass balance and length change, it was imposed to start the simulations without unwanted behavior at the initial stage.

We choose to let the glacier grow until the length indicated by the end moraine of the 19th century (4.62 km), as determined by lichenometric dating in the paleovalley (Boyarsky, 1978; Zolotarev, 1998; Petrakov et al., 2012), cf. Fig. 1. To obtain a steady state glacier, the multiyear mean mass balance profile for the 1967/68−2006/07 AD climate had to be increased by an additional mass balance perturbation of +1.12 m yr$^{-1}$ w.e., corresponding to an ELA lowering of 113 m. The steady state situation was then tested and verified as before (Sect. 2.3). It can be noted that modelled ice thickness around the maximum extent of the glacier in the considered model period went up to 173.4 m in the valley. Additionally, surface velocities were as high as 101.7 m yr$^{-1}$ near the ice falls of the Djantugan Plateau and up to 98.1 m yr$^{-1}$ in the valley downstream (Fig. 6d).

We furthermore chose to include a supraglacial debris cover in the initialization procedure. However, as can be deduced from the large lateral moraines in the Adylsu Valley (Fig. 1), the Djankuat Glacier used to export most of its debris to the margins in the historic period, rather than developing a supraglacial debris cover. Furthermore, debris sources from surrounding topography and melt-out processes were likely less widespread in the historic period because of the colder climate (i.e. the current exposed slopes were covered by the glacier itself and were more stable). Also, the fast-flowing nature of the paleo-glacier tongue in the valley (up to 100 m yr$^{-1}$ around 1752 AD, Fig. 6d) disfavours the accumulation of thick debris on the glacier surface. For this reason, supraglacial debris is believed to have been much less widespread prior to the observational period of 1967/68 AD, implying that the glacier was not very much influenced by debris cover in the historic period. Nevertheless, there is also indirect evidence for at least some supraglacial debris in the historic period from the presence of moraines in the valley (Fig. 1) and a photograph taken around 1930 showing some debris patches on the snout (Aleynikov et al., 2002b). It would be furthermore unrealistic to only introduce a debris cover in the model once the model approaches the start of the observations, as this would contradict the presence of moraines and the observation that

there already was an expanding debris cover during the first data collection in 1967/68 AD (Popovnin et al., 2015). However, because there is no direct evidence for the origin of the debris, it was chosen to include only melt-out processes in the model initialization, which implies that debris mass fluxes from surrounding topography are not incorporated in the initialization procedure (i.e. $F_{debris}^{input}$ = 0 m yr$^{-1}$ and $C_{debris}$ = 1.05 kg m$^{-3}$). With these values, the LIA steady state debris cover had a thickness of 0.64 m at the front and occupied a fractional area of ca. 8 % (ca. 0.331 km$^2$ of the 1752 AD glacier).

## 430    5.2 Evolution of the glacier from 1752 AD to present

To force the model in the historic period, climatic data at 3-hourly intervals are needed. Historic climatic datasets for Terskol weather station were therefore constructed using a multiproxy approach, including information from various weather stations in the area, such as Mestia, Pyatigorsk (approximately 100 km northeast from the glacier at 512 m elevation) and Mineralnye Vody (approximately 115 km northeast from the glacier at 321 m elevation). Additionally, historic data from the CRUTEM4

and CRU TS datasets, as well as from tree ring reconstructions for the broader Caucasus area, were used for the remaining uncovered data gaps since 1752 AD (D'Arrigo et al., 2001; Toucham et al., 2003; Akkemik et al., 2005; Akkemik and Aras, 2005; Griggs et al., 2007; Köse et al., 2011; Jones et al., 2012; Harris et al., 2014; Holobâcă et al., 2015; Martin-Benito et al., 2016; Dolgova, 2016). Data from the pre-observational period outside the Terskol time series were therefore averaged over all the available datasets to create a multiproxy mean time series, for which mean monthly temperatures and total

precipitation amounts were derived by matching the mean (corrected additively for temperature and multiplicatively for precipitation) and standard deviation of the overlapping part in the observed Terskol dataset (Table 2, e.g. Huss and Hock, 2015; Zekollari et al., 2019). To obtain a record with a 3-hourly temporal resolution, the data sequence for Terskol over which measurements with a 3-hourly interval are available is repeated into the past and future in order to maintain intra-daily and intra-annual variability in the data. These data were afterwards corrected for the monthly mean temperature and

precipitation amounts obtained in the previous step (Table 2). The reconstruction of temperature and precipitation clearly indicates a shift in the climatic conditions after 1752 AD. Especially during the last few decades, an accelerated warming trend has occurred, as the latest 10-year climatic interval exhibits a mean annual temperature anomaly of +0.5 °C compared to the 1981−2010 mean (Fig. 8a). This makes it the warmest period in the time series. For temperature, a clear sequence of colder and warmer intervals can be seen. Changes in precipitation show a sequence of drier and wetter periods (Fig. 8b).

After using the steady state glacier of 1752AD as an initial input feature for the time-dependent model, dynamic calibration is applied by incorporating artificial mass balance perturbations ($\Delta b_{(t)}$) into the model. This factor was not explicitly calculated but was instead derived and adjusted iteratively by a trial and error procedure. The obtained perturbations were then superimposed on the mass balance profile that was simulated with the climatic input, until the reconstructed glacier length sufficiently matched with the observed values (e.g. Oerlemans, 1997; Zekollari et al., 2014):

$$b_{a(x,t)} = b_{a(x,t)}^{SMB} + \Delta b_{(t)} \tag{18}$$

Here, $b_{a(x,t)}^{SMB}$ is the local surface mass balance simulated with the climatic datasets and $\Delta b_{(t)}$ is the artificial mass balance perturbation that was applied in the dynamic calibration procedure. Such a procedure is needed to counteract imperfections in the flow model, mass balance model and the climate forcing. The added value of this procedure is to ensure a current glacier state that matches the observed one, as the glacier is still responding to changes in past climate, geometry and dynamics. The procedure required a maximum additional mass balance perturbation of +0.5 m yr⁻¹ w.e. but which varies over time (Fig. 9b). Nevertheless, since the balance year 1967/68 AD, i.e. the year from which the mass balance model was calibrated, no additional perturbations were needed. It can thus be stated that the model performs well when forced with the observed Terskol climatic data, and that credibility can be assigned to the dynamic calibration procedure. It furthermore implies that future projections are no longer influenced by the corresponding artificial mass balance corrections, keeping in mind an e-folding length response time of ca. 31 years for the Djankuat Glacier (see Sect. 4).

The resulting mass balance series shows clear peaks around the 1870−80s, early 1900s, late 1910s, 1940s, 1970s and early 2000s AD, hereby coinciding with slightly colder and/or wetter periods in the climatic datasets (Fig. 9c). Clear minima in the mass balance series can be noted in the 1860s, 1890s, early 1910s, 1920s, late 1940s and in the 21th century, which agrees fairly well with earlier mass balance reconstructions of Djankuat (Dyurgerov and Popovnin, 1988; Fyodorov and Zalikhanov, 2018) and Garabashi glaciers on the Elbrus massif (Rototaeva et al., 2003; Dolgova et al., 2013). As the Djankuat Glacier reacted to these climatic perturbations, an almost continuous retreat since the 1850s AD had occurred, exhibiting some minor readvances or steady states as well. As was already discussed earlier, the past behaviour of the Djankuat Glacier is in line with the general observed trend for other Caucasian glaciers (Fig. 3). During the last several decades, however, the addition of a thickening layer of supraglacial debris on the snout aided to temporarily postpone rapid retreat and more or less maintain steady state conditions. Still, the glacier has lost a total length of 1.39 km at present-day compared to the start of the reconstruction in 1752 AD (-29.4 %). The reconstruction also shows that the total glacier area around 1752 AD decreased by 35.2 % when compared to the 2010 AD situation (an area of 4.147 against 2.688 km², see Figs. 6b and 9a). Moreover, evolution of glacier surface area matches nicely with observed values except for the outlier around 1983, which has to do with a migrating ice divide on the Djantugan Plateau (Fig. 9a).

A historic model run conducted with a 100 % clean-ice glacier, shown as an inset in Fig. 9a, revealed that debris played only a minor role prior to ca. 1980 AD, with length differences of only 20 to 40 m. By 2010 AD, however, the modelled length difference between a debris-free and debris-covered glacier already increased to 160 m (Fig. 9a).

## 6 Future glacier evolution to 2100 AD

### 6.1 Response to future climate forcing

Future projections of temperature and precipitation were obtained by a multi-model approach, using output from the Coupled Model Intercomparison Project Phase 5 (CMIP5) simulations (Taylor et al., 2012) for the grid cell closest to the Djankuat Glacier. Mean temperature and total precipitation amount at monthly resolution from 21 Global Circulation Models (GCMs)

for the RCP 2.6, RCP 4.5, RCP 6.0 and RCP 8.5 scenarios were used, based upon their availability (Table 2 and 3). The data

were downloaded for both historical runs (from 1981 AD) and for projections (until 2100 AD). Although the choice of ensemble member can largely influence the eventual results (e.g. Huss and Hock, 2015), we solely focus on the first realization, i.e. ensemble member r1i1p1. As with the historic climate datasets, climate data were scaled to match the mean and standard deviation of the Terskol meteorological station. Absolute GCM data were therefore at first scaled to anomalies with respect to the 1981−2010 reference values for each respective model, so that additive (temperature) and multiplicative

(precipitation) biases could be removed when matching to the past forcing. For each RCP, the monthly temperature and precipitation data were then averaged over all models, resulting in a multi-model mean time series. To account for year-to-year variability, the CMIP5 data were rescaled with respect to the standard deviation of the overlapping period for the observed Terskol data (e.g. Huss and Hock, 2015; Zekollari et al., 2019). As with the past, the observed 3-hourly Terskol data sequence was finally used to downscale the monthly data to the temporal resolution that suits the mass balance model.

Concerning debris cover evolution, the debris input location $x_{debris}$ and flux magnitude $F_{debris}^{input}$ were left unchanged. Consequently, once the contribution from $x_{debris}$ stops, either due to shrinkage of the surface width or rapid retreat beyond the input location, no additional debris source is released. Hence, only melt out from debris-loaded ice and supraglacial debris advection contribute to the evolution of the supraglacial debris cover afterwards. Later on, we will, however, conduct several experiments to determine the impact of potential additional debris sources from the surrounding topography on the

future glacier evolution (Sect. 6.2).

All scenarios exhibit a further increase of the temperature, which is most pronounced in the summer season. Projected precipitation, on the other hand, shows slightly decreasing values at annual resolution, but shows a tendency for a drier summer half year (April to September, AMJJAS) and a wetter winter half year (October to March, ONDJFM). By 2071−2010 AD, the mean AMJJAS temperature (total ONDJFM precipitation) anomalies with respect to the 1981−2010

period are +1.4°C (+0.1 %), +2.3°C (+3.7 %), +2.7°C, (+11.2 %) and +4.5°C (+11.7 %) for the RCP 2.6, RCP 4.5, RCP 6.0 and RCP 8.5 scenarios respectively (Figs. 10a and b). Additionally, also a future projection is made under a no change scenario, in which the last observed 10-year climatic interval (2009−2018 AD) is repeated with respect to its mean (corresponding to a AMJJAS mean temperature and a total ONDJFM precipitation amount anomaly of +0.5 °C and -11.0 % mm yr$^{-1}$ w.e. respectively).

All future scenarios agree to a rapid decline of the glacier length and surface area in the following decade, as a response to the significant warming since the late 1990s AD. By 2100 AD in the no change scenario, the total length and surface area of the glacier are projected to be 2370 m (-29.3 %) and 2.01 km² (-27.3 %), whereas the glacier front will be positioned at an elevation of 2844 m. It is thus clear that, at present day, the Djankuat Glacier is not in equilibrium with the current climatic conditions and hence will strive towards a new steady state with a much smaller surface area in the future. For the RCP 2.6,

4.5, 6.0 and 8.5 scenarios, the total glacier length further decreases to 1560 m (-52.2 %), 1250 m (-61.7 %), 1070 m (-67.2 %) and 510 m (-84.4 %) by 2100 AD respectively. Meanwhile, total glacier surface area decreases to 1.17 km² (-56.5 %),

0.71 km² (-73.6 %), 0.49 km² (-81.8 %) and 0.20 km² (-92.6 %) by 2100 AD respectively (Fig. 11a and b). As such, for the RCP 6.0 and 8.5 scenarios, the glacier retreats back as far as into the bedrock depression of the Djantugan Plateau.

With respect to total runoff volume changes and water resources management, the Djankuat Glacier is close to surpassing its
peak water discharge point, as the modelled annual glacier runoff reaches its maximum around 2020 AD (Fig. 11c). Hence, all RCP scenarios exhibit a further decline of the produced runoff volume into the future, which is in accordance with earlier work for this area (Huss and Hock, 2018; Hock et al., 2019). The actual course of runoff changes, however, is dependent upon the trade-off between remaining glacier surface area and magnitude of melt. As such, the RCP 8.5 scenario initially produces the highest melt and corresponding runoff volume. Later on, however, the 'no change' scenario yields the highest
runoff volumes due to the larger remaining glaciated area. It must also be noted that near the end of the modelling period, runoff volume temporarily stabilizes for the RCP 6.0 and RCP 8.5 scenarios. This process is related to the melting of the ice on the Djantugan Plateau, which then reinforces itself due to the mass balance-elevation feedback.

However, even under the most extreme RCP 8.5 scenario, the glacier would not completely disappear by the end of the modelling period. Despite accelerated melting of the high-elevation plateau because of the mass balance-elevation feedback,
a decreased climate sensitivity due to the steeper laterally averaged slopes in the upper glacier part, as well as the large ice thickness on the Djantugan Plateau (up to 200 m at present-day), prevent a complete disappearance by the end of the modelling period. It must furthermore be noted that the averaging of the future climatic data implies a reduction of spread. When, for example, the model was forced with the highest warming scenario of all CMIP5 models (i.e. the RCP 8.5 scenario of the GFDL-CM3 model, with mean AMJJAS temperature increase of +7.9 °C by 2071−2100 AD), the glacier will cease to
exist by 2086 AD.

### 6.2 Impact of supraglacial debris cover on glacier evolution

Despite present-day areas of visible clean ice on the tongue, a relatively steep slope below the ELA, relatively high ice velocities and a short response time, observations also show that the supraglacial debris cover on the Djankuat Glacier has significantly affected glacier geometry during the last several decades, as evident from the differential retreat of the snout
(Figs. 1 and 9a). Its importance for this specific glacier has also been demonstrated by e.g. Rezepkin and Popovnin (2018), who showed that the debris cover is believed to drastically affect the Djankuat Glacier in terms of its geometry and melting patterns. Debris input onto the Djankuat Glacier's surface due to mass fluxes from surrounding topography are furthermore expected to increase even further in the future (Popovnin et al., 2015; Rezepkin and Popovnin, 2018). To determine the potential effect of these additional debris sources onto the glacier surface, we performed additional experiments with varying
debris input location, debris input magnitude and time of the release of the debris source from the surrounding topography. We repeated the procedure used in Sect. 2.5, but indicate a 'debris reference scenario', in which a second debris mass flux is initiated from $x_{debris} = x_{ELA}$ at $t_{debris} = 2035$ with a magnitude of $F_{debris}^{input} = 1.5$ m yr⁻¹. For $x_{ELA}$, the average position of the ELA was calculated during a window of ±15 years surrounding $t_{debris}$ in the 'no additional debris scenario' (Sect. 6.1),

which hence varies for each climatic scenario. We therefore choose to not initiate debris fluxes from positions above the ELA, due to the neglect of englacial pathways in our debris model (see Sect. 2.5). We then let one of these three variables change, while keeping the other two at their original value of the 'reference situation'. As such, the debris input location $x_{debris}$ was changed to 80 %, 60 % and 40 % of the distance between $x_{ELA}$ and $x_L$ (further downstream), the time of release $t_{debris}$ to 2045, 2055 and 2065, and at last the magnitude of the debris flux $F_{debris}^{input}$ to 0.75, 2.25 and 3.0 m yr$^{-1}$. It must be noted that the values of these parameters are arbitrary, as the exact location, time and magnitude of future debris sources cannot be predicted. By assessing a range of possible values for each of these parameters, we encompass various potential future scenarios in order to account for the high uncertainty regarding these parameters. Figure 12 shows the impact of these variables (rows) on the future length of the Djankuat Glacier under different climatic scenarios (columns). The black lines indicate the scenario where no additional debris source is released in the future. The other lines are for experiments that include an additional future debris source from the surrounding topography for varying values of the earlier mentioned debris-related parameters. It is clear that the addition of an increasingly widespread debris cover dampens glacier retreat. It should however be noted that the effects on glacier length are not immediate, as it takes some time for the debris to be advected to the terminus after its initiation at time $t_{debris}$.

The effect of the timing of the source release is straightforward: the earlier the debris mass flux is released, the larger the extension of the glacier by the year 2100 AD, as the melt-reducing effect starts earlier in time. The main decisive factor here is the efficient debris advection towards the terminus, because flow velocities are larger in 2035 AD compared to 2050, 2065 and 2080 AD (Fig. 12). The magnitude of the debris input flux $F_{debris}^{input}$ is another crucial parameter determining the length extension of the Djankuat Glacier in the future period. It is, hence, obvious that a higher flux magnitude will contribute more efficiently to a higher debris growth rate. This enhanced effect is a direct consequence of the implementation of Eq. 14, where the debris-related melt reduction depends on the debris thickness (Fig. 12). Concerning the debris input location, results suggest that the closer the input source is located to the terminus, the longer the extension of the glacier will be compared to the situation without an additional debris source. This makes sense, as the time that it takes for the supraglacial debris to be advected to the front is shorter for down-glacier input locations. Hence, the debris cover will be able to apply its melt-reducing effect much earlier in time, as well as much further down-glacier in space on a still relatively long glacier. Again, the effects on glacier length are not immediate, as it takes some time for the debris to be advected to the terminus.

The effect of climatic conditions on debris-related melt reduction and its impact on glacier geometry is twofold. Initially, the melt-reducing effect increases with higher temperature, as can be seen in the case of the no change, RCP 2.6 and RCP 4.5 scenarios. This can be related to the fact that a higher temperature will increase the melt-out of material from debris-loaded ice, whereas also decreased flow velocities prevent sufficient discharge and allow the debris to thicken quickly up-glacier (Fig. 12). Moreover, the distances between the input point and the glacier front at the time of source release decrease with increasing temperature, whereas also retreat rates are relatively larger for higher temperatures. This allows the relatively thick debris to encounter the glacier front much earlier in time. At last, it is important to note that for the same melt reduction

factor $f_{debris}$, the absolute reduction of the ablation amount will be higher when the initial value of the ablation is high. However, for the RCP 6.0 and RCP 8.5 scenarios, the impact of the supraglacial debris cover on the glacier decreases again. Here, a counteracting effect occurs as temperatures rise even further, because the risk of rapid loss of debris-covered area increases. This can be related to either the breaking of the glacier into several fragments where areas of 'dead ice' prevent proper connectivity between the main glacier body and the glacier front, or because the front is too close to (or has already passed) the debris source by the time it is released. Finally, the accelerated shrinkage also favors foreland deposition instead of debris accumulation due to frontal retreat, as well as the loss of proper connectivity between the debris source and the main glacier body at the debris input location (Eq. 13, Fig. 12).

**7 Conclusion**

In this study, a coupled ice flow−mass balance−supraglacial debris cover model was used to simulate the response of the Djankuat Glacier to past, present and future climatic changes between 1752 and 2100 AD. We conducted, for the first time, explicit time-dependent modelling of a Caucasian glacier, including an extended and physically based subroutine related to supraglacial debris cover evolution that was not yet integrated in time-dependent numerical flow line models. As it turns out, the Djankuat Glacier has been retreating almost continuously since the 1850s AD, with some minor steady states or readvances during periods with clusters of colder and/or wetter conditions. The model reconstructed the observed retreat fairly well but required additional mass balance perturbations up to a maximum of +0.5 m yr⁻¹ w.e., which were applied iteratively via dynamic calibration. However, since the start of the calibration period in the balance year 1967/68 AD, no artificial mass balance perturbations were needed, ensuring proper model calibration and credibility.

The future behaviour of the glacier is determined by corresponding changes in air temperature, precipitation and supraglacial debris cover. A temperature increase of 1 °C can only be compensated by a precipitation increase of ca. 35 %, which is not indicated by future climatic projections in the study area. Hence, all scenarios agree to a rapid decline during the following decade, as a response to the accelerating warming since the 1990s AD. Even after considering constant present- day climatic conditions, the glacier will shrink drastically by ca. 30 % of its current length and surface area by 2100 AD, indicating the imbalance between the current glacier geometry and the present climate. However, none of the future scenarios cause a total disappearance by the end of the modelling period. Nevertheless, the glacier will retreat most drastically (ca. -93 % of its current surface area) under the RCP 8.5 scenario, as even the thick ice on the high elevations of the Djantugan Plateau will be affected by significant melting. Although the glacier is close to surpassing its peak water discharge point, the modelled temporal evolution of total runoff volumes indicates that, in particular the melting of ice on these higher parts of the glacier in higher-temperature scenarios, temporarily stabilizes runoff near the end of the modelling period due to the mass balance-elevation feedback.

The presence of a supraglacial debris cover is shown to significantly affect glacier geometry during the modelling period. Hence, the effect of debris-related melt reduction on the eventual glacier length by 2100 AD is dependent upon the trade-off

between the growth rate of the total supraglacial debris mass, the efficiency of down-glacier advection of supraglacial debris,
the glacier retreat rate, the connectivity between the debris source and the main glacier, and finally the distance between the
front and the input location at the time of source release. It turns out that debris-related effects are highest when either debris
thickness and area are large, or when melt-reducing effects start earlier in time and/or more down-glacier in space in a
relatively warm climate. However, it must be noted that for some of the conducted experiments, the addition of an extra
debris source did not (significantly) influence the glacier's geometry. As such, when temperatures increase even further,
potential inhibiting effects of too rapid shrinkage are to be considered. Hence, accelerated frontal retreat, disrupted debris
discharge and/or connectivity issues at the debris input location may prevent the establishment of a proper melt-reducing
effect.

*Code and data availability*. The model code was written in MATLAB_R2019a. A coupled ice flow-supraglacial debris cover
model for the Djankuat Glacier, that was used as the basis for this research, can be found and downloaded from*:
https://github.com/yoniv1/Djankuat_glacier_model*. A subfolder with the climatic datasets is made available through the
same repository (doi: http://doi.org/10.5281/zenodo.3934612).

*Author contribution*. Y.V. created the climatic datasets, constructed and calibrated the numerical model, performed the
numerical simulations and wrote the manuscript. P.H. proposed the main conceptual ideas and outlines, helped design and
implement the research, provided guidance in interpreting the results and improved the manuscript throughout the entire
process. O.R. and V.V.P. contributed by making glacier field work possible, providing numerous datasets and improving the
manuscript with their knowledge and years of experience concerning the Djankuat Glacier.

*Competing interests*. The authors declare that they have no conflict of interest.

*Acknowledgements*. The contribution of V.V. Popovnin and O. Rybak was supported by the Russian Foundation for Basic
Research, grant RFBR No 18-05-00420a: "The latest evolutionary tendencies in water and ice resources of the glaciers in the
Caucasus". The authors furthermore like to thank all researchers Vladimir M. Fyodorov, Olga Solomina, Dario Martin-
Benito, Ekaterina Dolgova, Dimitry A. Petrakov, Iulian H. Holobaca and Pavel A. Toropov, who provided their climate
reconstruction data and knowledge, and hence helped to improve the quality of the historic datasets greatly. We also thank
the reviewers Loris Compagno, Ann Rowan and Fabien Maussion and the editor Evgeny A. Podolskiy for their helpful
comments, which significantly improved the quality of the manuscript.

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

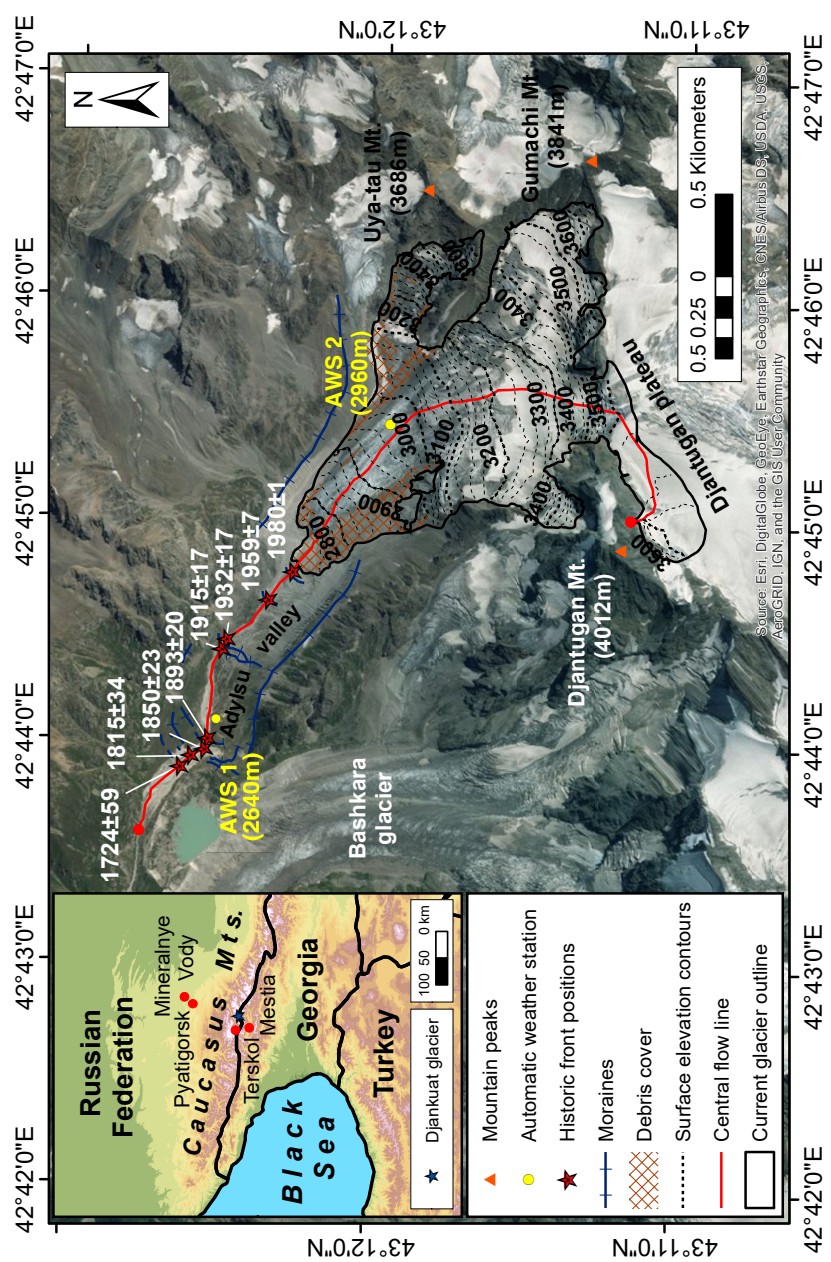

**Figure 1.** Satellite image of the Djankuat Glacier for the year 2010 AD, showing the most important features in the study area.

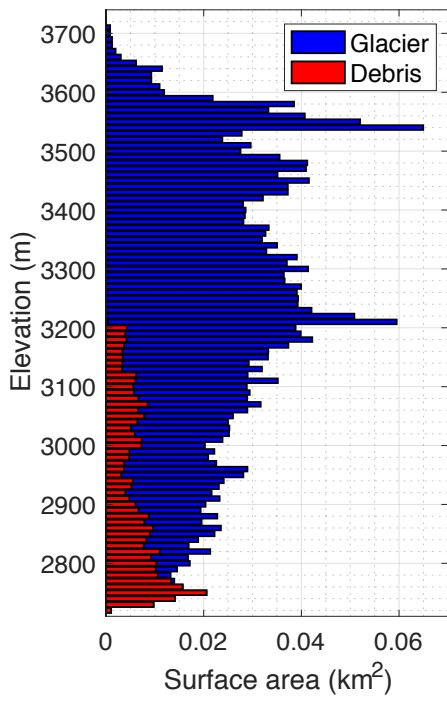

**Figure 2.** The Djankuat Glacier's surface (blue) and debris covered area (red) for 2010 AD conditions as shown by the area-
elevation distribution using 10-m bins. Hypsometric data are derived from the DEM and manual digitalization of the
supraglacial debris cover using satellite imagery in Fig. 1.

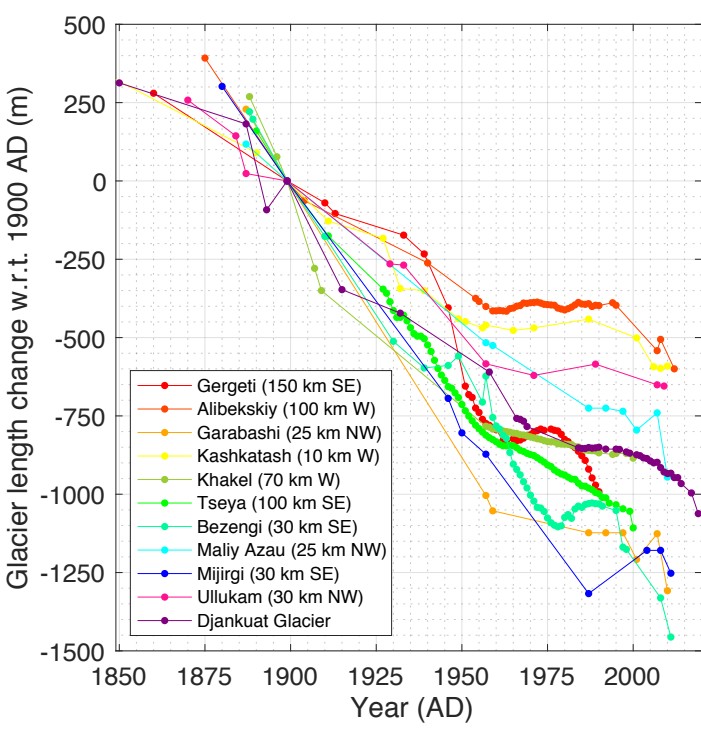

**Figure 3.** Historic length variations of the Djankuat Glacier compared to other glaciers in the Caucasus (Solomina et al., 2016; WGMS, 2018). Approximate distances and direction to the Djankuat Glacier are indicated.

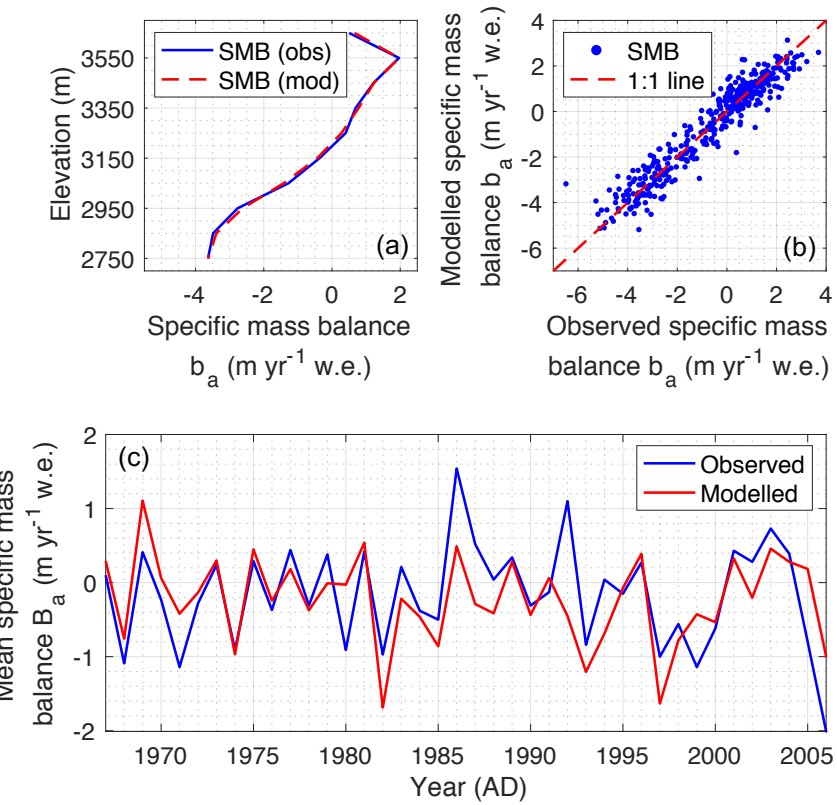

**Figure 4.** Calibrated mass balance model of the Djankuat Glacier for fixed geometry, showing the observed and modelled (a) mass balance-elevation profile for the 1967/68−2006/07 period, (b) local annual surface mass balances $b_a$ for the 1967/68−2006/07 period and (c) modelled and observed mean specific mass balance $B_a$ since the start of the monitoring period. Observed mass balance data are retrieved from Popovnin and Naruse (2005) and WGMS (2018).

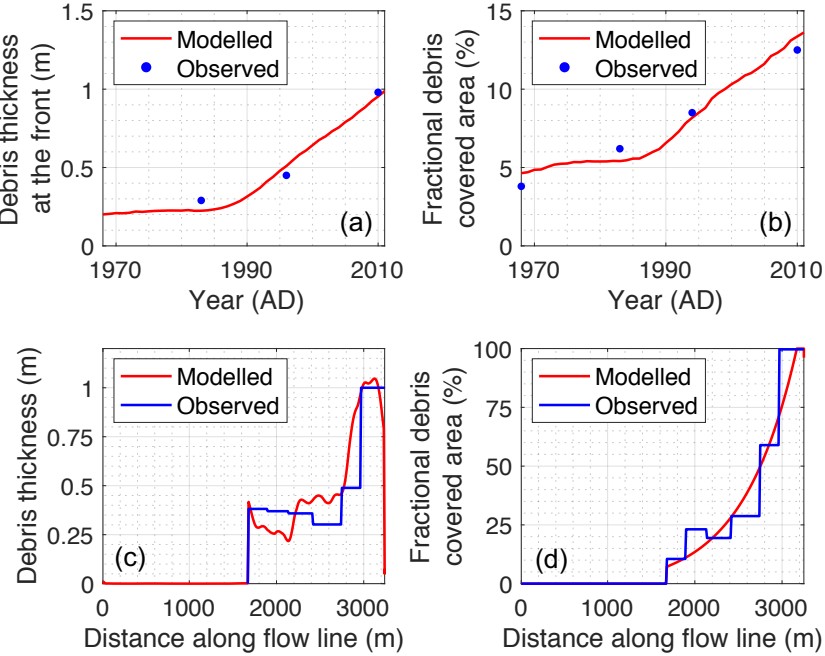

**Figure 5.** Calibrated supraglacial debris cover model for the Djankuat Glacier, showing the observed and modelled temporal evolution of (a) debris thickness at the front and (b) the glacier-wide fractional debris covered area, as well as observed and modelled (c) debris thickness and (d) debris covered area along the flow line for 2010 AD conditions. Observed data from (a), (b) and (c) are from Popovnin et al. (2015), whereas the observed debris covered area in (d) was derived by manually digitizing debris-covered patches along the flow line using 2010 AD satellite imagery in Fig. 1.

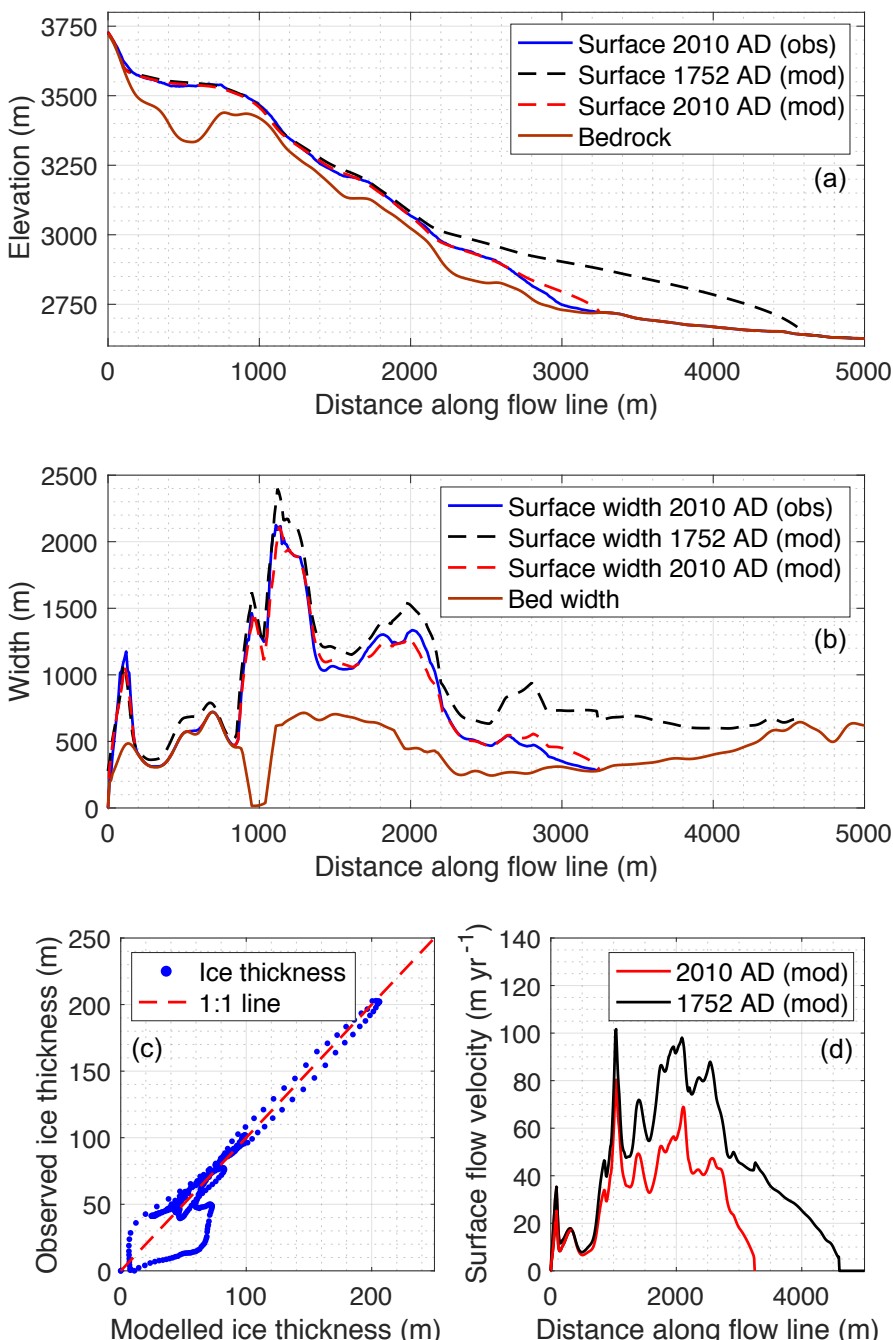

**Figure 6.** Calibrated flow model, showing (a) the observed and modelled bedrock and surface elevation and (b) bed and surface width for the current (2010 AD) and initial state (1752 AD), (c) modelled vs. observed ice thickness for 2010 AD conditions and (d) current (2010 AD) and initial (1752 AD) surface flow velocity along the flow line.


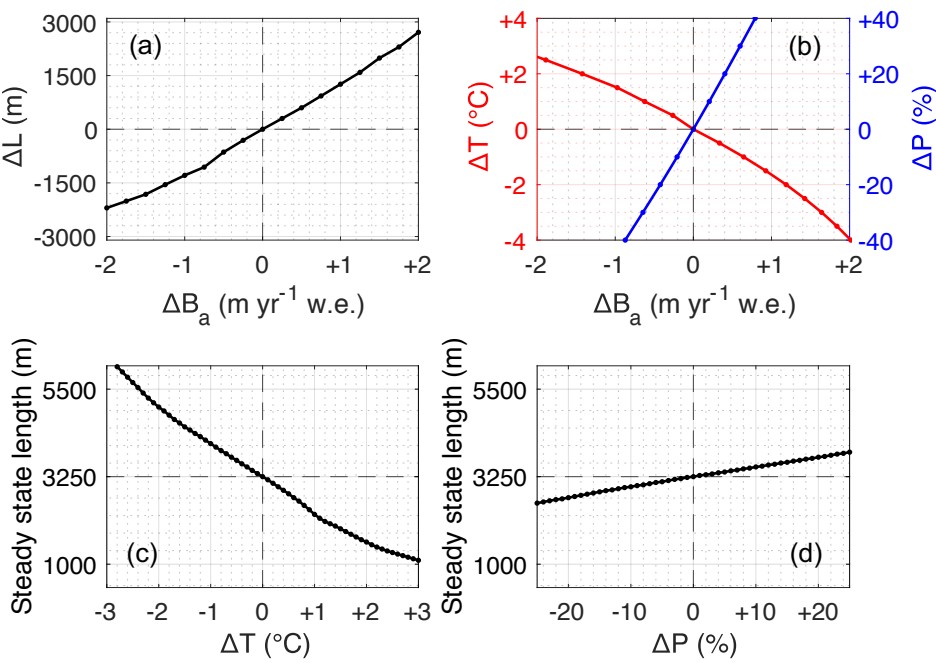

**Figure 7**. Sensitivity of the Djankuat Glacier showing (a) sensitivity of the glacier steady state length (ΔL) to mass balance perturbations ($\Delta B_a$), (b) sensitivity of the mass balance to temperature (ΔT) and precipitation (ΔP) changes for a fixed present-day glacier geometry, (c) sensitivity of the steady state glacier length to temperature changes, and (d) the same for precipitation changes. All perturbations are with respect to the 1967/68−2006/07 AD reference climate (2.5 °C and 980.7 mm yr$^{-1}$ w.e.), and with respect to a steady state glacier with present-day length.


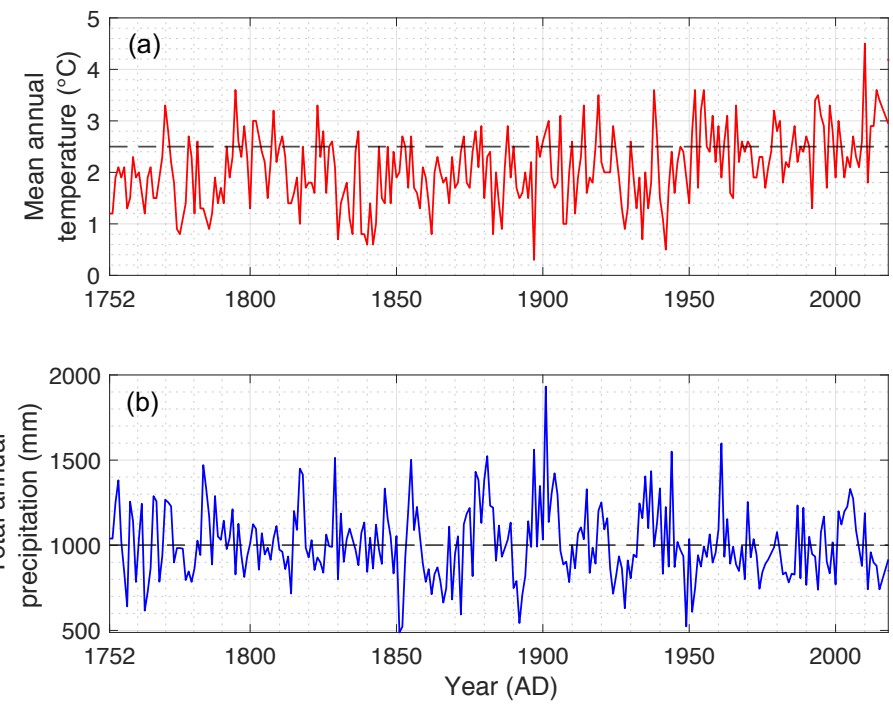

**Figure 8.** Reconstructed and observed evolution of (a) mean annual temperature and (b) total precipitation amounts for Terskol weather station, based upon proxy data (tree ring reconstructions) and measurements from nearby weather stations (Mestia, Pyatigorsk and Mineralnye Vody). The dashed horizontal line represents the 1981−2010 annual reference values (2.6 °C and 1001.1 mm yr$^{-1}$ w.e.). We refer to the text and Table 2 for more details.


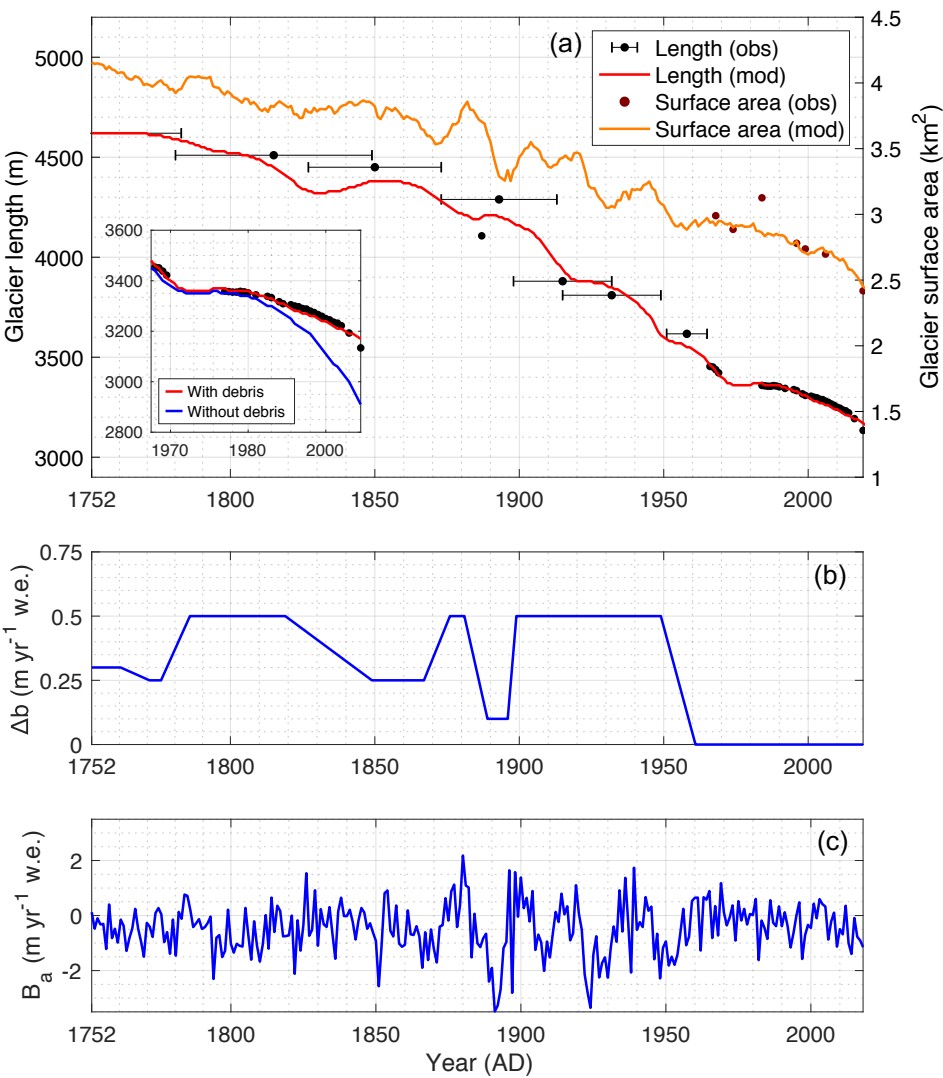

**Figure 9.** Historic variations of the (a) modelled and observed glacier length and surface area of the Djankuat Glacier until 2017 AD, (b) additional mass balance perturbations $\Delta b$ used in the dynamic calibration procedure and (c) reconstructed time series of the total annual mass balance $B_a$ of the Djankuat Glacier with changing geometry. Observed length variations are derived from lichenometric dating of moraines in the paleovalley, historic documents, field measurements and/or recent satellite imagery (Boyarsky, 1978; Zolotarev, 1998; Petrakov et al., 2012; WGMS, 2018). An additional model run for a 100% clean ice glacier was conducted, which is shown in the box in (a).

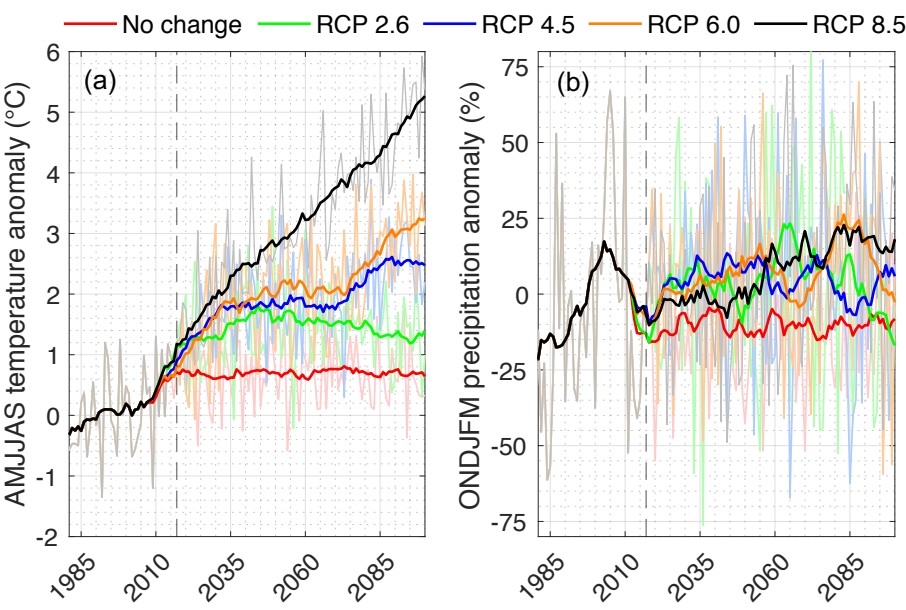

**Figure 10**. Projected future (a) AMJJAS temperature and (b) ONDJFM precipitation changes for Terskol, as compared to the 1981−2010 reference, for different RCP scenarios until 2100 AD. Thin coloured lines represent annual values, thicker lines represent 15-yr moving means. The dashed vertical line represents the present (i.e. 2017, the most recent year of glaciological observations).

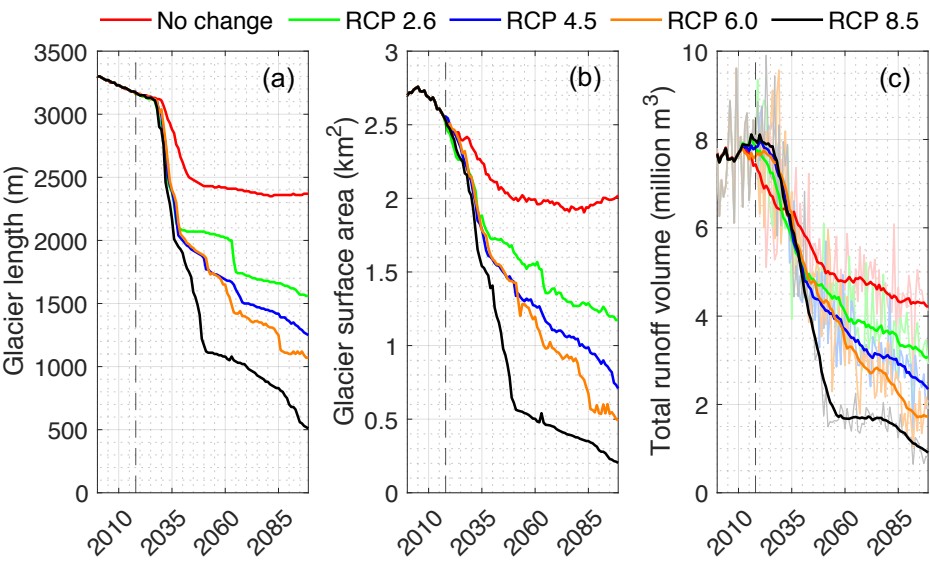


**Figure 11**. Modelled (a) glacier length, (b) glacier surface area, and (c) total annual runoff volume of the Djankuat Glacier for different RCP scenarios until 2100 AD. In (c), the thin lines represent annual values, while the thicker lines represent 15-yr moving average. The dashed vertical line denotes the present (i.e. 2017, the most recent year of glaciological observations).


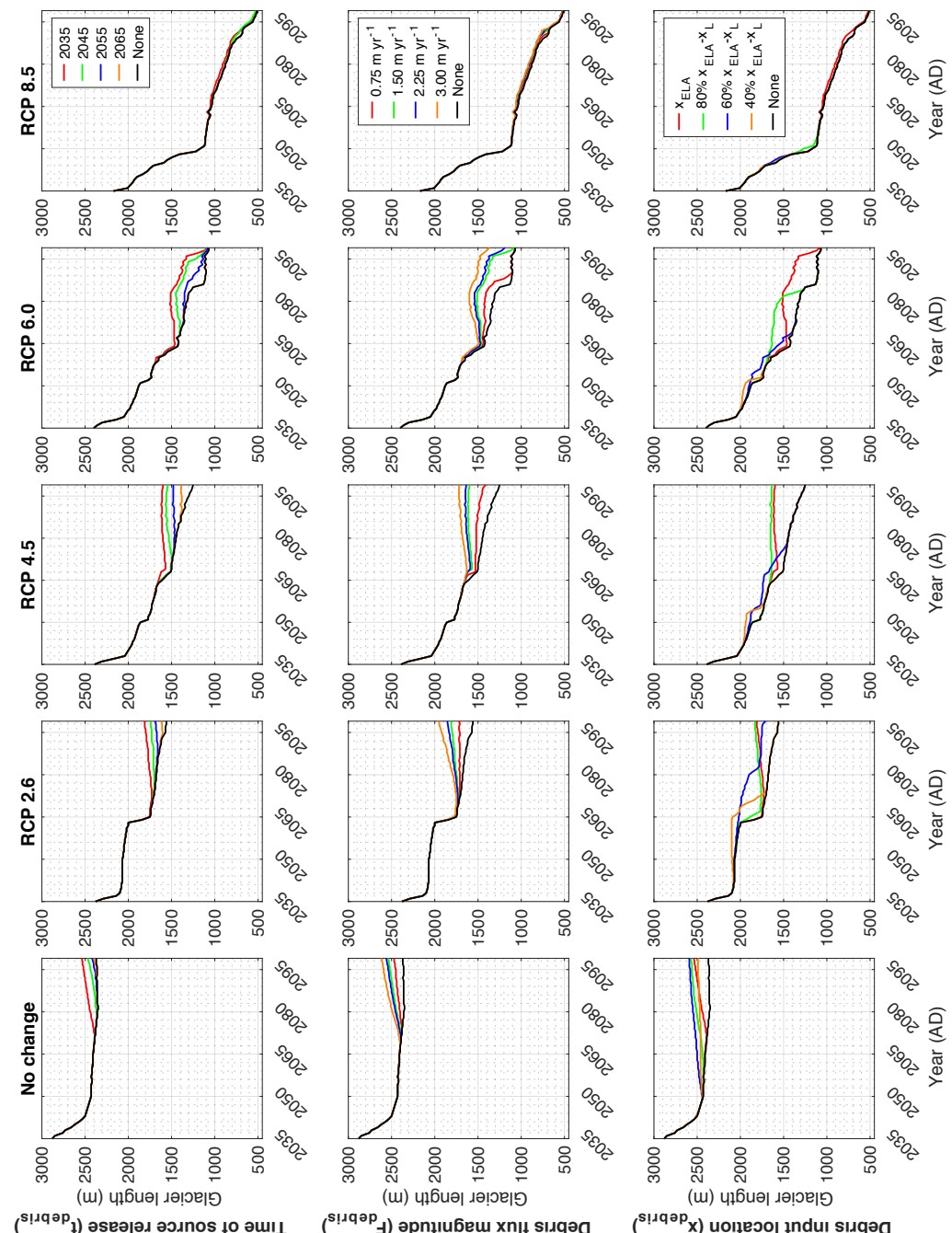

**Figure 12.** Impact of debris input location $x_{debris}$, time of release of the debris source $t_{debris}$ and debris flux magnitude $F_{debris}^{input}$ (rows) on the future length evolution of the Djankuat Glacier under different climatic scenarios (columns) after 2035 AD.

**Table 1.** Variables, constants and their units used in the model. The hyphen (–) denotes that the value is not a constant.

| Variable | Symbol | Value | Unit | Variable | Symbol | Value | Unit |
|---|---|---|---|---|---|---|---|
| | | | | **Supraglacial debris cover model** | | | |
| Timestep debris model | $\Delta t$ | 0.01 | yr | Spatial resolution debris model | $\Delta x$ | 10 | m |
| Characteristic debris thickness | $H^*_{debris}$ | 1.15 | m | Debris melt-reduction factor | $f_{debris}$ | - | - |
| Debris thickness | $H_{debris}$ | - | m | Growth factor debris area | $G_A$ | - | $yr^{-1}$ |
| Debris-covered area | $A_{debris}$ | - | $km^2$ | Englacial debris concentration | $C_{debris}$ | 1.05 | $kg\,m^{-3}$ |
| Debris cover porosity | $\phi_{debris}$ | 0.43 | - | Debris rock density | $\rho_{debris}$ | 2600 | $kg\,m^{-3}$ |
| In/output of debris w.r.t. the glacier surface | $I_{debris}$ | - | $m\,yr^{-1}$ | Input flux to the glacier surface at input location | $F^{input}_{debris}$ | 1.60 | $m\,yr^{-1}$ |
| Debris input location | $x_{debris}$ | 1680 | m | Deposition flux into the foreland | $F^{x=L+1}_{debris}$ | - | $m\,yr^{-1}$ |
| Foreland deposition rate of debris at terminus | $F^{x=L}_{debris}$ | - | $m\,yr^{-1}$ | Distance along flow line | $x$ | - | m |
| Time of release of debris source | $t_{debris}$ | 1958 | yr | Constant for strength of debris foreland deposition | $c_L$ | 1 | $m^{-1}$ |
| Distance to the front | $D_L$ | - | m | Average debris thickness of first 30 grid points | $H^{front}_{debris}$ | - | m |
| | | | | **Mass balance model** | | | |
| Timestep mass balance model | $\Delta t$ | 3 | hours | Spatial resolution mass balance model | $\Delta x$ | 10 | m |
| Surface elevation | $h$ | - | m | Fraction of diffuse solar radiation | $f_{dif}$ | 0.50 | - |
| Elevation of Terskol weather station | $h_{Terskol}$ | 2141 | m | Fraction of direct solar radiation | $f_{dir}$ | 0.50 | - |
| Elevation of AWS on Djankuat | $h_{AWS}$ | 2960 | m | Angle of incidence | $\theta$ | - | ° |
| Elevation of AWS in Adylsu Valley | $h_{Adylsu}$ | 2640 | m | Solar elevation angle | $\theta_e$ | - | ° |
| Horizontal precipitation enhancement between Terskol and Adylsu Valley | $f_e$ | 1.5 | - | Solar zenith angle | $\theta_z$ | - | ° |
| Snow redistribution factor | $f_{red}$ | - | - | Fractional cloud cover | $f_{cl}$ | - | - |
| Precipitation ratio between glacier and Adylsu Valley | $P_{scale}$ | - | - | Snow depth | $d_{snow}$ | - | m w.e. |
| Threshold air temperature for rain-snow distinction | $T_{tresh}$ | 2.0 | °C | Incoming extra-terrestrial shortwave radiation at the TOA | $S_{\downarrow(TOA)}$ | - | $W\,m^2$ |
| Temperature lapse rate summer | $\gamma_{T(S)}$ | -0.0067 | $°C\,m^{-1}$ | Characteristic snow depth | $d^*_{snow}$ | 0.011 | m w.e. |
| Temperature lapse rate winter | $\gamma_{T(W)}$ | -0.0049 | $°C\,m^{-1}$ | Outflow of retained melt water from snow | $W_{snow}$ | - | m w.e. |
| Precipitation lapse rate over glacier | $\gamma_P$ | 0.0023 | $m\,yr^{-1}\,m^{-1}$ | Liquid snow store | $w_{snow}$ | - | m w.e. |
| Net energy flux at glacier surface | $\Psi_0$ | - | $W\,m^2$ | Snowpack retention capacity | $\eta_s$ | 0.34 | - |
| Albedo for ice | $\alpha_{ice}$ | 0.22 | - | Latent heat of fusion | $L_m$ | 334 000 | $J\,kg^{-1}$ |
| Albedo for snow | $\alpha_{snow}$ | 0.79 | - | Density of water | $\rho_w$ | 1 000 | $kg\,m^{-3}$ |
| Intercept $\Psi_0(T_{air})$ | $c_0$ | -39.0 | $W\,m^{-2}$ | Threshold temperature $\Psi_0(T_{air})$ | $T_{break}$ | 0.0 | °C |
| Slope $\Psi_0(T_{air})$ | $c_1$ | 13.0 | $W\,m^{-2}\,°C^{-1}$ | Atmospheric transmissivity | $\tau$ | 0.53 | - |
| Critical slope for loss due to redistribution | $s_{crit}$ | 25 | ° | Melt production from snow/ice | $M$ | - | $m\,s^{-1}$ w.e. |
| Local annual (or specific) surface mass balance | $b_a$ | - | $m\,yr^{-1}$ w.e. | Total annual (or mean specific) mass balance | $B_a$ | - | $m\,yr^{-1}$ w.e. |
| | | | | **Ice flow model** | | | |
| Timestep flow model | $\Delta t$ | 0.0005 | yr | Spatial resolution flow model | $\Delta x$ | 10 | m |
| Distance along flowline (x-direction) | $x$ | - | m | Ice thickness | $H$ | - | m |
| Vertically averaged horizontal velocity | $\overline{u}$ | - | $m\,yr^{-1}$ | Surface elevation | $h$ | - | m |
| Velocity related to internal deformation | $\overline{u}_d$ | - | $m\,yr^{-1}$ | Effective slope related to lateral valley wall angles | $\mu$ | - | - |
| Velocity related to basal sliding | $u_s$ | - | $m\,yr^{-1}$ | Ice density | $\rho_i$ | 917 | $kg\,m^{-3}$ |
| Surface velocity | $u_{sfc}$ | - | $m\,yr^{-1}$ | Gravitational acceleration | $g$ | 9.81 | $m\,s^{-2}$ |
| Ice volume flux | $F_{ice}$ | - | $m^3\,yr^{-1}$ | Flow parameter related to internal deformation | $f_d$ | $6.5 * 10^{-17}$ | $Pa^{-3}\,yr^{-1}$ |
| Width (glacier surface) | $W_{sfc}$ | - | m | Flow parameter related to basal sliding | $f_s$ | $3.25 * 10^{-13}$ | $Pa^{-3}\,m^2\,yr^{-1}$ |
| Width (glacier bed) | $W_0$ | - | m | Glacier length | $L$ | - | m |

**Table 2.** Input data used for the Terskol climate reconstruction (1752−2100 AD).

| Meteorological parameter | Source | Temporal resolution | Extent of dataset | Applied correction |
|---|---|---|---|---|
| Precipitation | Proxy data (D'Arrigo et al., 2001; Toucham et al., 2003; Akkemik et al., 2005; Akkemik & Aras, 2005; Griggs et al., 2007; Köse et al., 2011; Martin-Benito et al., 2016) | Variable | 1752−present | (a) Use multi-proxy / multi-model mean approach.<br><br>(b) Bias correction for precipitation (multiplicative) biases and year-to-year variability (standard deviation), see e.g. Huss and Hock (2015) and Zekollari et al. (2019).<br><br>(c) Convert to 3-hourly values by using the observed Terskol data sequence as base but corrected for monthly amounts derived before. |
| | CRU TS v4.02 dataset (Harris et al., 2014) | Monthly | 1901−present | |
| | Pyatigorsk weather station | Daily | 1934−1997 | |
| | Mestia weather station | Monthly | 1961−2010 | |
| | Terskol weather station | 3-hourly, monthly | 1977−present (gap 1990−1997) | |
| | Mineralnye Vody weather station | Daily | 1938−present | |
| | CMIP5 simulations (Taylor et al., 2012) | Monthly | present−2100 | |
| Temperature | Proxy data (Holobaca & Pop, 2015; Dolgova et al., 2017) | Variable | 1752−present | (a) Use multi-proxy / multi-model mean approach.<br><br>(b) Bias correction for temperature (additive) biases and year-to-year variability (standard deviation), see e.g. Huss and Hock (2015) and Zekollari et al. (2019).<br><br>(c) Convert to 3-hourly values by using the observed Terskol data sequence as base but corrected for monthly amounts derived before. |
| Temperature | CRUTEM4 v4.6.0.0 dataset (Jones et al., 2012) | Monthly | 1850−present | |
| Temperature | Mineralnye Vody weather station | Daily | 1938−present | |
| Temperature | Mestia weather station | Monthly | 1961−2010 | |
| Temperature | Terskol weather station | 3-hourly, monthly | 1977−present (gap 1990−1997) | |
| Temperature | CMIP5 simulations (Taylor et al., 2012) | Monthly | present−2100 | |




**Table 3**. CMIP5 climate models used for the Terskol climate projections (2019−2100 AD).

| Model | Spatial resolution | RCP 2.6 | RCP 4.5 | RCP 6.0 | RCP 8.5 |
|---|---|---|---|---|---|
| BCC-CSM1-1-M | 2.81°×2.81° | X | X | X | |
| INMCM4 | 1.50°×2.00° | | X | | X |
| ACCESS1-3 | 1.25°×1.88° | X | X | | X |
| CNRM-CM5 | 1.41°×1.41° | X | X | | X |
| IPSL-CM5A-LR | 1.90°×3.75° | | X | X | X |
| IPSL-CM5B-LR | 1.90°×3.75° | X | X | | X |
| MPI-ESM-MR | 1.88°×1.88° | X | X | | X |
| GFDL-ESM2G | 2.00°×2.00° | X | X | X | X |
| GISS-E2-R | 2.00°×2.50° | | X | X | X |
| HadGEM2-CC | 1.25°×1.88° | | X | | X |
| ACCESS1-0 | 1.25°×1.88° | | X | | X |
| BCC-CSM1-1 | 2.81°×2.81° | X | X | X | X |
| BNU-ESM | 2.81°×2.81° | X | X | | X |
| IPSL-CM5A-MR | 1.25°×2.50° | X | X | X | X |
| MPI-ESM-LR | 1.88°×1.88° | X | X | | X |
| NorESM1-M | 1.88°×1.88° | X | X | X | X |
| CMCC-CMS | 3.75°×3.75° | | X | | X |
| GFDL-CM3 | 2.00°×2.50° | X | X | | X |
| GFDL-ESM2M | 2.00°×2.50° | X | X | X | X |
| GISS-E2-R-CC | 2.00°×2.50° | | X | | X |
| HadGEM2-ES | 1.25°×1.88° | X | X | X | |
