# Peer review of "Modelling the evolution of Djankuat Glacier, North Caucasus, from 1752 until 2100 AD"

_The Cryosphere, 2019_

## Referee Comment (RC1) · Loris Compagno (Referee) · 4 Mar 2020

**Summary**

Y. Verhaegen et al. used a combined ice flow-mass balance-debris evolution model to simulate the behavior of Djankuat Glacier, North Caucasus from 1752 to 2100.
This is the first study where Djankuat Glacier was modeled for such a long-time period, as well as using a model which explicitly takes into account debris evolution.
In its current form, the study has 3 minor limitations related to the way future climate is dealt with, to missing informations regarding a mass balance perturbation factor, and the sensitivity analysis.
I think that these limitations can be improved relatively easily and, therefore, I support the publication of this manuscript after the required improvements (**minor revisions**).

**General Comments**

1) **Future climate**
   The surface mass balance model was forced with climatic observations in the past, and with CMIP5 climate scenarios for the future (l. 410-419). However, some important information seems to be missing:
   (i) Which climate models did you use (model name, institute, resolution, …)?
   (ii) Did you applied a de-biasing procedure to accommodate the future climate projections with the past climatic dataset (e.g. Huss, 2015)? Such a procedure is often needed to avoid sudden changes in temperature and precipitation between the past climate dataset and the future climate projections.
   (iii) Why did you use a linear trend (l. 413-415) for the future temperature and precipitation and not the trend (and variability) proposed by the CMIP5 data? This virtually discards any CMIP5 information between now and the end of the century…

2) **Mass balance perturbation**
   In the manuscript, a mass balance perturbation is used as a tuning factor, so that model results agree with observations (l. 330, 390, 493, fig. 9 and fig. 11). However, how this perturbation factor is calculated and applied is not well explained. Here additional information are absolutely needed so that the reader can understand what this factor is and how it is meant.

3) **Model sensitivity**
   l. 330-338 show a sensitivity analysis. However, also here many important informations are missing.
   (i) Are these experiments done starting by a glacier steady state? If yes, during which time period?
   (ii) Your results show how much the glacier length changes for each degree (°C) of warming, using the unit 'm/°C', (line 335).  Over what temperature-range can the glacier response be expected to be linear? It seems easy to imagine that the topography of the glacier and its bedrock play a role, since they are not homogeneous and thus influence the glacier response depending on the glacier's position?

**Line-by-line comments**

l. 14: better to say already in the abstract which future climate data you used.

l. 24-25: this sentence needs some references.

l. 40: change '.,' to ','.

l. 44-45: are you referring to the whole Caucasian region?

l. 65: you cannot use one glacier as representative for a whole area (Huss, 2008). However, at lines 78-79 it becomes clearer what you meant. So, please reformulate.

l. 89: give a number for 'higher elevations'

l. 101: (i) it is not clear which mean annual temperature you are referring to (mean temperature of the 1981-2010 period?). (ii) it is not clear what 'here' is referring to.

l. 106-107: (i) you mentioned two places and then you say that you used only one automatic weather station (AWS). Did you used the same AWS in the two places? (ii) 'was installed' –> can you add from when to when? This information is especially important if you used only one AWS for two places.

l. 114: maybe add 'glacier' before 'top', so that it becomes 100% clear.

l. 117-119: sorry, I cannot follow this sentence. Can you maybe reformulate it?

l. 121 (eq1) The way Eq. 1 is cast looks somewhat unusual to me. Can you maybe add a reference where the derivation can be looked up? Or add the derivation in the manuscript?

l. 134: spell out 'FTCS'

l. 141: remove 'specific', since it is the glacier wide balance here

l. 145: Is there one value of ACC for the whole glacier, is it evaluated along the central flowline, or is there some sort of spatial grid playing a role?

l.149: (Oct-Mar) add also the day, or whether the beginning or end of the month are meant.

l. 154: add link to table 1 already after 'gamma_p'

l. 144-159: not super clear to me, especially how exactly all these factors are derived.

l. 167: and alpha? Add that alpha is the albedo.

l. 180: about which 'tilt' are you speaking? Is the AWS station tilted?

l. 189: 'more or less' - please use a synonym.

l. 189: 'Table 1' - It took me quite a lot to find values that you were referring to. Can't it simply be added to the text?

l. 192: 'plotted' – Is this the correct word? With 'plotted' I expect a Figure…

l. 200-205: Is the implicit assumption that $C_{debris}$ is homogeneous within the entire glacier body? Since that's unlikely to be true, the assumption should at least be discussed.

l. 208: 'at 1680 m from the highest point' – where is this point? Maybe show this location in Fig.1.

l. 209-211: the choice of stopping the debris input flux at a given glacier width sounds rather arbitrary. Also the fact that the debris input location $x_{debris}$ is fixed in time (and not moving) causes some doubts. Both points seem to merit some discussion.

l. 215: (eq. 13) The variable '$t_{debris}$' is not introduced.

l. 225: can you give some more details about the relationship which was found?

l. 228: how is the debris-area growth factor $G_A$ 'updated yearly'? One should be pointed at eq. 17 at this stage.

l. 234: you took into account the melting reduction effect of debris, but what's about the melting enhancement effect of thin debris (e.g. Østream, 1959)? Add some discussion about this.

l. 247: 'this time period' – maybe re-state the time period.

l. 251, 254: use always the same unit.

l. 258: what's the meaning of 'between 0.18 and +/- 0.6 m'.

l. 261: 'second a' → 'a second'

l. 267: can you give some numbers about the 'snow redistribution by wind/avalanche'?

l. 270-273: Did you validate the model with the same data which were used also to calibrate the model? If not, specify which data you used. If yes, isn't there a different, independent dataset which can be used for model validation?

l. 280: I don't understand what $t_{debris}$ exactly is (cf. l. 215).

l. 297: 'the bed was slightly adjusted' – how? Can you give some more details?

l. 326: is the volume change a yearly volume change? If yes correct the unit.

l. 331: please correct the unit/make it consistent with the rest of the manuscript.

l. 332: can you add a reference or some details about the 'e-folding length response time'?

l. 334 and 335: well, I imagine that these numbers depend a lot on the topography (see general comments)?

l. 330-338: the sensitivity experiments need some more details on how they are done (see general comments).

l. 336-337: 'an acceptable accuracy' – please give a number.

l. 381-387: in my opinion, all these sentences can be reduced into one or two sentences.

l. 390-395: How is this mass balance perturbation obtained (see main comments)?

l. 411-413: Can you give some more details about these data? Did you use global circulation models (GCMs) or regional climate models (RCMs) ? Which model did you used (all GCMs and RCMs have specific names, realizations, institutions…)

l. 413-415: Why did you use a linear temperature and precipitation evolution? Why not using the transient evolutions of the climate models? Moreover: did you applied a de-biasing approach between past climate dataset and the future climate projections? (see main comments)

l. 417: '+7.1 °C' compared to when? Is this number a temperature difference between two periods or a temperature mean? If it is a temperature mean, please report also the value of the first period, or the difference.

l. 455-456: Are these values arbitrary? That seems fine but if they are, better add a sentence explaining why these values are taken and from where.

l. 503: '-80%' – of area? Of volume? Or of length?

**Comments to figures and tables**

Fig.1: the scale bar is bizarre. Why not using 0, 0.25, 0.5, 1 km instead of 0.5, 0.25, 0, 0.5 km?

Fig.2: Would be nice to have a little map showing where these other glaciers are, or at least their distance to Djankutan Glacier.

Fig.4: (i) I may have missed it, but what is causing the modelled MB gradient to 'flip' at the highest elevations (>3550 m) in Fig. 4a?
(ii) Also here the units are different than in the main text.

Fig.5a+b: (i) Would be interesting to see longer-term evolution of these values as well. (ii) What's causing the sharp bend around 1985? Is it the switch to very negative MBs

Fig. 10: (i) add labels for temperature and precipitation, (ii) say what "w.r.t." is and (iii) say what the black line is.

Fig. 11: What is the black line?

Fig. 12: Can the volume be plotted as well? That seems an important quantity.

Fig. 13: (i) unit in the y-axis (/year?), (ii) say in the caption what 'present' is for this study (what year?).

Fig. 14: Why are the impacts visible only after ca. 2050?

General comment about figures: The 'YYYY/YY' format is pretty distracting. Better use 'YYYY'

Table 1: Can you explain what the '-' means?

**References**

Huss, M., Bauder, A., Funk, M. & Hock, R. Determination of the seasonal mass balance of four Alpine glaciers since 1865. *J. Geophys. Res.* **113**, F01015 (2008).

Huss, M. & Hock, R. A new model for global glacier change and sea-level rise. *Frontiers in Earth Science* **3**, (2015).

Østrem, G. Ice Melting under a Thin Layer of Moraine, and the Existence of Ice Cores in Moraine Ridges. *Geografiska Annaler* **41**, 228–230 (1959).

---

## Referee Comment (RC2) · Ann Rowan (Referee) · 20 Mar 2020

Verhaegen and co-authors present a numerical modelling study of Djankaut Glacier, investigating the evolution of this glacier from the Little Ice Age (1752 CE) to 2100 CE. This is a small, 3 km2, glacier that has some surface debris. The focus of the paper is therefore on the impact of supraglacial debris on mass balance and how this changes over time. I have several major concerns about the work undertaken and the content of the manuscript which are detailed below. There are also numerous minor points for improvement and I have mentioned some of these in my review. My main concern the relevance of using a debris transport–mass balance model to a glacier with a thin and discontinuous debris layer, particularly when glacier length and area metrics are used to evaluate glacier change. As a modelling study of a WGMS benchmark

glacier there is much value in exploring how it will change and quantifying sensitivities to climate. However, the work in its present form lacks the scope and rigour required for publication in The Cryosphere, and the presentation of the manuscript would benefit from additional work, including adding a Discussion section.

Major comments 1. Evaluation of glacier change in terms of terminus position and glacier area. We know that debris-covered glaciers have a different dynamic response to climate warming based on remote sensing observations and numerical modelling, which shows that they lose the majority of mass by surface lowering rather than terminus recession and area reduction. Therefore, the latter metrics that are useful for clean-ice glaciers are poor indicators of the behaviour of a debris-covered glacier. My main concern with the present study is that the debris-cover model is unnecessary given the characteristics of Djankaut Glacier (e.g. large areas of visible clean ice on the tongue, steep slope below the ELA, high velocities, large changes in length and area over several decades, short response time) and therefore introduces a bias to the results. It would be valuable to demonstrate the difference between simulations with and without the debris-cover model to evaluate its impact on glacier change and if the observed change can be replicated without this additional calculation. It appears that this information is contained in Fig. 14, which shows future glacier evolution under different climatic forcings, but this is not discussed in the text and the figure is difficult to interpret; it appears that the debris has no impact on glacier length change until the second half of the century.

2. Value of sub-debris melt calculation. In relation to my point above, I have two concerns about the debris-cover model; (a) the gradient of the exponential function used to scale sub-debris melt is steep using H*debris = 1.15 m (see review Fig. 1), and (b) the thickness of debris on the glacier is similar to the critical thickness observed on debris-covered glaciers elsewhere and therefore likely to both enhance and reduce ablation across the tongue. The glacier model accounts for the impact of supraglacial debris by reducing mass balance, a valid assumption beneath debris that is thicker

than a critical thickness of about 0.1–0.2 m. An exponential function is used to reduce ablation with debris thickness. However, images of the present day glacier including Fig. 1 and data presented in the manuscript (Fig. 5a) illustrate that the debris thickness at the terminus is ~1.0 m in 2010 and was <0.5 m before 1990. As debris thickness decreases rapidly upglacier (Fig. 5c), is the same scaling is assumed then most of the debris layer was <0.25 m thick before 1990 and therefore close to the critical thickness. For such thin and discontinuous debris layers, there is likely to be little reduction in ablation due to insulation by the debris layer (the exponential function used here will only reduce sub-debris melt by <20% compared from the clean-ice value – see review Fig. 1) and instead an enhancement of ablation due to the reduction in albedo of debris-covered ice compared to clean-ice surfaces. The model does include an albedo term but does not use this to adjust for the impact of debris on ablation.

2. Evolution of the debris layer. As observed in Fig. 5, the debris layer has thickened by a factor of 2–3 over the last 20 years. Djankaut Glacier is steep, fast-flowing and thinly debris-covered over a section of its ablation area, and based on this geometry and the presence of large ice-marginal moraines it seems likely that during the LIA and subsequently, the glacier exported the majority of its debris to its margins rather than developing a supraglacial layer. Therefore, the assumption in the spin up simulation that the glacier is debris covered (Section 5.1) may not hold. However, from Fig. 14 it appears that the debris layer has no impact on glacier change until about 1970 CE.

2. Lack of Discussion. The manuscript organisation is somewhat unconventional. After the Introduction, Methods and Model Description, there are four Results sections (not named as such) followed by the Conclusions. There is very limited discussion of the context of results and their interpretation, and no dedicated section for this.

Minor comments by line number Line 13. "retreat" see major comment 1 about terminus recession versus surface lowering.

L15–16. The change in glacier length and area stated here are not meaningful unless

the initial length and area are also given, or these are stated as % change.

L24–25. Vague statement.

L39. Use of "significantly" should be reserved to its precise statistical meaning, whereas here it is used for emphasis and could be replaced with "dramatically" or in this sentence the meaning would be the same if this word was removed.

L45–47. What is the glacierised area and debris-covered area in the Caucasus in km2? This is needed to indicate the context suggested in this statement.

L53. Citations to previous modelling studies of debris-covered glaciers. Please note that Rowan et al. (2015) did not use a simple parameterisation of the impact of debris on mass balance as stated here, but instead made a dynamic simulation of the feedbacks between ice flow, debris transport and mass balance using a higher-order ice flow model. The statement ending in line 64 is therefore incorrect, as previous studies have taken this approach. A citation to Wirbel et al. (2017) should also be included: Wirbel, A., Jarosch, A.H. and Nicholson, L., 2018. Modelling debris transport within glaciers by advection in a full-Stokes ice flow model. The Cryosphere, 12(1), pp.189-204.

L73. State glacier area here.

L96–98. Use metres for debris thickness values here to be consistent with the rest of the text.

L101. "Mean annual air temperature", and "+" is not needed before the values.

L110. Explain what you mean by "1.5D" or stick with "1D" to indicate a flow line calculation. L112. Do you mean 2D rather than "3D", i.e. a matrix calculation?

L224. Give value for H*debris, from Table 1, the value used after tuning was 1.15 m, which results in the steep curve mentioned in Major Comment 1. Also it is not clear as written here how this model compared to that presented in Anderson and

Anderson (2016) as mentioned in the Introduction, which used a hyperbolic rather than exponential function to scale sub-debris melt; h*/(h*+hdebris) their Eq. 3 with h* of 0.065 m.

L258 and 260. Unclear as written. What is the meaning of "±" before the values given for H*debris? Do these values range from –0.6 to 0.6 m?

L259 and elsewhere. One of the key references for a previous application of this model to Djankaut Glacier is Rybak et al. (2018), which is cited to justify parameter choices and to give detail about the model. However, this document is difficult to locate and appears to only be available in Russian. I was not able to use this reference to collect information about the model. At Line 259 the citation here is incorrect, as "Rybak (2018)" is not in the reference list.

L363. All the models have different time steps; 3-hourly for the mass balance model, ∼4 hourly for the ice flow model and ∼4 days for the debris transport model. How are the integrated, and what impact do these time steps have on the result when the response time is ∼30 years?

L455–459. What evidence is there for the choice of debris input parameters?

L490. Incorrect statement, see comment on line 53 above.

L508–518. Here and elsewhere, although the written text is generally clear and free of typographic errors, the writing style is rather vague and qualitative, using large lists of variables/controls without indicating their importance, and the meaning can be difficult to follow. The manuscript would benefit from editing to enable clearer, more precise statements to present the study and its results.

The code and data used are described as available on request from the author. I believe the Cryosphere now requires these to be open access in a repository.
* * *
[Figure]

**Fig. 1.**

---

## Referee Comment (RC3) · Fabien Maussion (Referee) · 30 Mar 2020

**1    General comments**

In this paper, Verhaegen and co-authors use a flowline model to simulate the evolution of Djankuat Glacier from 1752 to 2100. I am late with this review because of the current circumstances, my apologies. This review takes the comments from two other reviewers into account, and I would also like to acknowledge the help of Matthias Dusch (University of Innsbruck), with whom I discussed the presented manuscript since it is relevant for our own research.

The manuscript by Verhaegen and co-authors is well written and comprehensive. A

substantial effort has been made to describe the models and data used, but it must be said that given the number of parameters and data sources, it is very difficult to make a thorough review of all model aspects. One could argue over certain model details (such as the choice of a 3 hr time step for the MB model for example), but I think that it is not very relevant here. Instead, I would like to make four main comments:

- **model validation**: at several places in the manuscript, the authors say that the model was validated, for example: "*It can thus be stated that the model performs well and underwent a successful validation to within acceptable accuracy*". I argue that a model is validated when its capacity to reproduce the "unseen" is assessed (past and future evolution, or unobserved variables). A model is useful when model predictions are associated with an uncertainty estimate. As it is now, the model has a very large number of free parameters which are calibrated to match observations almost perfectly. Per design, the study does not allow validation with independent or out-of-sample data (e.g. cross-validation). I don't think that it is possible to change this aspect of the study at this stage, but I would like to see the problem of model uncertainty and over-calibration discussed in the manuscript, and the statement that the model has been successfully "validated" should be changed to "calibrated". I think that the consequences of parameter equifinality are most likely to be seen in the sensitivity experiments of the debris cover parameterization and the future projections.

- **added value of the past simulations**: the model is dynamically tuned to fit observed length changes, with a time varying bias parameter. I am aware that this has been done before (and will be done in the future), but I have to ask: in the end, what is the added value of such a simulation? What do we learn from it, that we didn't already know from length change observations alone? What are the implications of the dynamic parameterization for the future projections?

- **debris cover parameterization**: in my opinion, the true added value of this study

lies in the coupling of a debris parameterization with the flowline model. I think it would add great value to the manuscript to extend the sensitivity analyses to the past glacier simulation as well (which, as it stands, is of very limited usefulness). How is the past glacier evolution changed by the inclusion of debris cover? In order not to make this paper even longer, I would suggest to remove Fig. 7 to 9, which are quite qualitative.

• **Code and data availability**: you write: "*the refined debris cover implementation can be used for comparable glacier models in future research*". I agree! But it would be considerably more useful if the code and data used in this study would be made freely available under a proper software license and in a public repository. Platforms like zenodo.org will preserve the version of the model as it is at the time of this publication. And it will create a DOI to make it citable for future research. See TC's data policy: https://www.the-cryosphere.net/about/data_policy.html

Altogether, it is still a nice study and a useful addition to the literature.

**2 Specific comments**

**Abstract L10** : I would prefer not to use the term "1.5D". I never understood what the "0.5" is referring to: the widths? The vertically integrated velocity? Should a 2D SIA model then be called a 2.5D model? I think that a "SIA flowline model" is explicit enough.

**Figure 1** : If possible, indicate the location of AWS Adylsu Valley

**Figure 2** : It is misleading to compare length changes like this, because they all have a different zero baseline. It would be much better to plot them all as relative length change since year X (e.g. since year 2000).

**Figure 6a and chapter 5.1** : Could you elaborate on why the 1752 steady-state glacier has a longer and thicker tongue while the ice thickness above 3000 m a.s.l. is more or less equal to the 2009 glacier?

**Figure 10 and 11c** : I assume the black lines are rolling means. Please specify years.

**L 106** : Please specify which data period and parameters are available at these stations.

**L 134** : The time step is fixed at 0.0005 years. Did any stability considerations or tests go into this choice?

**L 147, 179, 183** : Which time period was used?

**L 149** : Was the winter temperature lapse rate solely chosen based on the reported ELA temperature by WGMS (2018) or was AWS data used as well?

**L 150, 179, 191** : It is often not clear if data from the AWS Djankuat or the AWS Adylsu Valley is used.

**L 150, 250** : Is the precipitation scaled to match one of these AWS? If yes which one?

**L 162 (Eq. 7)** : It might be worth noting that this melt term is only one part of the total runoff of the mass-balance model and that the rest is derived in the next chapter.

**L 187** : Can you please specify how the fractional cloud cover is parametrized?

**L 192** : From that sentence I would expect a Figure similar to Figure A1 of Giesen and Oerlemans (2010).

**L 369** : "At first, data from the pre-observational period ..."

**L 370** : Terskol time period is already specified in Table 2

**L 373** : How was the available data repeated into the past? By just copying the entire time period? Shuffling of individual days/months/years? Were any sensitivity tests made in that regard?

**L 374** : Terskol time period is already specified in Table 2 and line 370.

**L 378-388** : This paragraph (and also L 396-400) with the listing of different dates and periods is a bit cumbersome to read. Maybe it would be better to indicate these periods in the anyway mentioned Fig. 10 and be more concise in the text.

**L 393-395 (and L494-495)** : The mass-balance and debris cover models were calibrated for the period 1967-2007 with the use of multiple tuning parameters to fit the observed surface mass-balance. The fact that no further dynamic calibration via mass-balance perturbations was necessary for this period can not lead to conclusions about the model performance and accuracy.

**Future glacier evolution** : Like other reviewers, I do not understand how the GCM climate is used in this study. Why is the linear change necessary, why not applying the GCMs delta T and delta P directly?

---

## Author Comment (AC1) · 10 Jul 2020

RC 1 – Loris Compagno (rebuttal by Verhaegen et al. – tc-2019-312)

Thank you for your helpful comments. In the text below, reviewer comments are indicated with colored background, our replies are in plain text and changes to the manuscript are put in italic.

**Response to general comments**

**General comment 1**

>> Future climate: The surface mass balance model was forced with climatic observations in the past, and with CMIP5 climate scenarios for the future (l. 410-419). However, some important information seems to be missing:

(i) Which climate models did you use (model name, institute, resolution, …)? (ii) Did you applied a de-biasing procedure to accommodate the future climate projections with the past climatic dataset (e.g. Huss & Hock, 2015)? Such a procedure is often needed to avoid sudden changes in temperature/precipitation between the past climate dataset and the future climate projections. (iii) Why did you use a linear trend (l. 413-415) for the future temperature and precipitation and not the trend (and variability) proposed by the CMIP5 data? This virtually discards any CMIP5 information between now and the end of the century…

We acknowledge that our original approach to derive future climate projections used a simplified linear approximation for the 21$^{th}$ century based on average CMIP5 model output. We have now recreated the future climate forcing directly from available CMIP5 models for the grid cell closest to Djankuat Glacier. In this way we encompass the variability captured by the CMIP5 models. We also applied a de-biasing procedure to match the future climate forcing with the past, both concerning the trend and the variability. The following text is now used to describe the future climate forcing and replaces lines 411-419:

*"Future projections of temperature and precipitation were obtained by a multi-model approach, using output from the Coupled Model Intercomparison Project Phase 5 (CMIP5) simulations (Taylor et al., 2012) for the grid cell closest to the Djankuat Glacier. Mean temperature and total precipitation amount at monthly resolution from 21 Global Circulation Models (GCMs) for the RCP 2.6, RCP 4.5, RCP 6.0 and RCP 8.5 scenarios were used, based upon their availability (Table 2 and 3). The data were downloaded for both historical runs (from 1981 AD) and for projections (until 2100 AD). Although the choice of ensemble member can largely influence the eventual results (e.g. Huss and Hock, 2015), we solely focus on the first realization, i.e. ensemble member r1i1p1. Absolute data were at first scaled to anomalies with respect to the 1981−2010 reference values for each respective model, so that additive (temperature) and multiplicative (precipitation) biases could be removed when matching to the past forcing. For each RCP, the monthly temperature and precipitation data were then averaged over all models, resulting in a multi-model mean time series. To account for year-to-year variability, the CMIP5 data were at last rescaled with respect to the standard deviation of the overlapping period for the observed Terskol data (e.g. Huss and Hock, 2015; Zekollari et al., 2019). As with the past simulations, the observed 3-hourly Terskol data sequence was finally used to downscale the monthly data to the temporal resolution that suits the mass balance model.*

*All scenarios exhibit a further increase of the temperature, which is most pronounced in the summer season. Projected precipitation, on the other hand, shows slightly decreasing values at annual resolution, but shows a tendency for a drier summer half year (April to September, AMJJAS) and a wetter winter half year (October to March, ONDJFM). By 2071−2010 AD, the mean AMJJAS temperature (total ONDJFM precipitation) anomalies with respect to the 1981−2010 period are +1.4°C (+0.1 %), +2.3°C (+3.7 %), +2.7°C, (+11.2 %) and +4.5°C (+11.7 %) for the RCP 2.6, RCP 4.5, RCP 6.0 and RCP 8.5 scenarios respectively (Fig. 10a and b). Additionally, also a future projection is made under a no change scenario, in which the last observed 10-year climatic interval (2009−2018 AD) is repeated with respect to its mean*

*(corresponding to a AMJJAS mean temperature and a total ONDJFM precipitation amount of 9.2°C and 373.3 mm yr⁻¹ w.e. respectively)."*

A figure in which these results are summarized is added as Figure 10 in the updated manuscript, and replaces Table 2 of the original manuscript:

[Figure]

*New Figure 10 in the updated manuscript. Projected future (a) AMJJAS temperature and (b) ONDJFM precipitation changes for Terskol, as compared to the 1981−2010 reference, for different RCP scenarios until 2100 AD. Thin colored lines represent annual values, thicker lines represent 15-yr moving means. The dashed vertical line represents the present (i.e. 2017, the most recent year of glaciological observations).*

A new Table 3 was added to the updated manuscript, indicating which climate models were selected to reconstruct the future forcing:

*New Table 3 in the updated paper. CMIP5 climate models used to reconstruct the future forcing (2019−2100 AD).*

| Model | Spatial resolution | RCP 2.6 | RCP 4.5 | RCP 6.0 | RCP 8.5 |
|---|---|---|---|---|---|
| BCC-CSM1-1-M | 2.81∘×2.81∘ | X | X | X | |
| INMCM4 | 1.50∘×2.00∘ | | X | | X |
| ACCESS1-3 | 1.25∘×1.88∘ | X | X | | X |
| CNRM-CM5 | 1.41∘×1.41∘ | X | X | | X |
| IPSL-CM5A-LR | 1.90∘×3.75∘ | | X | X | X |
| IPSL-CM5B-LR | 1.90∘×3.75∘ | X | X | | X |
| MPI-ESM-MR | 1.88∘×1.88∘ | X | X | | X |
| GFDL-ESM2G | 2.00∘×2.00∘ | X | X | X | X |
| GISS-E2-R | 2.00∘×2.50∘ | | X | X | X |
| HadGEM2-CC | 1.25∘×1.88∘ | | X | | X |
| ACCESS1-0 | 1.25∘×1.88∘ | | X | | X |
| BCC-CSM1-1 | 2.81∘×2.81∘ | X | X | X | X |
| BNU-ESM | 2.81∘×2.81∘ | X | X | | X |
| IPSL-CM5A-MR | 1.25∘×2.50∘ | X | X | X | X |
| MPI-ESM-LR | 1.88∘×1.88∘ | X | X | | X |
| NorESM1-M | 1.88∘×1.88∘ | X | X | X | X |
| CMCC-CMS | 3.75∘×3.75∘ | | X | | X |
| GFDL-CM3 | 2.00∘×2.50∘ | X | X | | X |
| GFDL-ESM2M | 2.00∘×2.50∘ | X | X | X | X |
| GISS-E2-R-CC | 2.00∘×2.50∘ | | X | | X |
| HadGEM2-ES | 1.25∘×1.88∘ | X | X | X | |

A small section was added related to the application of the most extreme scenario (Line 449):

*"The averaging of the future climatic data implies a reduction of spread. When, for example, the model was forced with the highest warming scenario of all CMIP5 models (i.e. the RCP 8.5 scenario of the GFDL-CM3 model, with mean AMJJAS temperature increase of +7.9°C by 2071−2100 AD), the glacier will cease to exist by 2086 AD."*

The resulting projections of future glacier geometry are included in the comment to "Fig. 12" below.

**General comment 2**

>> Mass balance perturbation: In the manuscript, a mass balance perturbation is used as a tuning factor, so that model results agree with observations (l. 330, 390, 493, fig. 9 and fig. 11). However, how this perturbation factor is calculated and applied is not well explained. Here additional information are absolutely needed so that the reader can understand what this factor is and how it is meant.

The mass balance perturbation $\Delta B_a$, used in the dynamic calibration procedure, was not explicitly calculated but was instead derived by a trial and error procedure. In this regard, the values for $\Delta B_a$ were iteratively adjusted to calibrate historic length variations of the glacier. These artificial mass balance perturbations were therefore superimposed on the mass balance profile that was simulated with the climatic input, until modelled and observed historic length coincided. The process is now clarified in the updated paper by explicitly stating how the procedure was applied (Lines 389-392 of the original paper):

*"…applied by incorporating artificial mass balance perturbations ($\Delta B_a$) into the model. This factor was not explicitly calculated but was instead derived and adjusted iteratively by a trial and error procedure. The obtained perturbations were then superimposed on the mass balance profile that was simulated with the climatic input, until the reconstructed glacier length sufficiently matched with the observed values (e.g. Oerlemans, 1997; Zekollari et al., 2014):*

$B_a = B_{a(SMB)} + \Delta B_a,$

*Here, $B_{a(SMB)}$ is the mass balance simulated with the climatic datasets and $\Delta B_a$ is the artificial mass balance perturbation that was applied in the dynamic calibration procedure."*

For more information, we refer to general comment 2 of RC 3.

**General comment 3**

>> Model sensitivity: l. 330-338 show a sensitivity analysis. However, also here many important informations are missing. (i) Are these experiments done starting by a glacier steady state? If yes, during which time period? (ii) Your results show how much the glacier length changes for each degree (°C) of warming, using the unit 'm/°C', (line 335). Over what temperature-range can the glacier response be expected to be linear? It seems easy to imagine that the topography of the glacier and its bedrock play a role, since they are not homogeneous and thus influence the glacier response depending on the glacier's position?

(i) All sensitivity experiments were conducted with respect to a steady state glacier with present-day length (i.e. a length of 3260 meter). We clarified this in the paper, and Lines 330-331 now read:

*"Some basic sensitivity tests were conducted with the flow model, which all initially started from a steady state glacier with present-day geometry. Perturbed mass balance profiles ($\Delta B_a$, in steps of 0.25 m $yr^{-1}$ w.e.) were then used as forcing to the model, until a new steady state was reached. As such, a relationship with a slight deviation from linear was found between the steady state length and $\Delta B_a$, exhibiting a value for $\partial L/\partial B_a$ of ca. 1100 and 1355 m (m $yr^{-1}$ w.e.) $^{-1}$ for negative and positive perturbations respectively (Fig. 7a)."*

(ii) We acknowledge that glacier geometry plays a decisive role in determining the climate sensitivity (m/°C) of the glacier. In this regard, sensitivity to temperature changes shows a linear behavior (815 m°$C^{-1}$) for temperature perturbations between -1.4 and +0.7°C (with respect to the 1967/68−2006/07 reference climate). Outside this range, the climate sensitivity slightly deviates from that linear trend, see Fig. 7c below. As suggested, the bed topography is key to explain this

behavior, as the glacier front is located on significantly steeper terrain for higher temperatures. Clarification has also been added to the text, as Lines 334-335 now read:

*"To assess the climate and glacier sensitivity for equilibrium, mass balance profiles were furthermore altered by temperature and precipitation perturbations within the -3 to +3°C and -25 % to +25 % range respectively (as compared to the 1967/68−2006/07 reference values). Sensitivity of steady state length to temperature changes ($\partial L / \partial T_{air}$) shows a linear behavior (815 m°C$^{-1}$) for perturbations between -1.4 and +0.7°C, but is modelled to vary between 400 and 1400 m°C$^{-1}$ when assessed over the entire range (Fig. 7c). The glacier sensitivity depends largely upon geometry and increases (decreases) for more negative (positive) mass balance perturbations, predominantly due to the flatter (steeper) terrain. The sensitivity also peaks around a temperature perturbation of +1°C, i.e. when the glacier front is positioned at the transition between the broad accumulation area and the narrower snout (ca. x = 2300 m on the flow line). Also the non-linear nature of the temperature-mass balance relationship (Fig. 7b) triggers a deviation from linear behavior. Consequently, the change in forcing needed for a retreat from 2 to 1 km is nearly twice as large as for a retreat from 4 to 3 km. For precipitation the sensitivity is more or less constant for a value of 250 m 10 %$^{-1}$ (Fig. 7d)."*

We present the new Fig. 7 as a replacement to the old Fig. 7-9 (in Sect. 4 of the original paper):

[Figure]

*"New Figure 7 in the updated manuscript. Sensitivity of the Djankuat Glacier showing (a) sensitivity of the glacier steady state length to mass balance perturbations (ΔBₐ), (b) sensitivity of the mass balance to temperature (ΔT) and precipitation (ΔP) changes for a fixed present-day glacier geometry, (c) sensitivity of the steady state glacier length to temperature changes, and (d) the same for precipitation changes. All perturbations are with respect to the 1967/68−2006/07 AD reference climate (2.5°C and 980.7 mm yr$^{-1}$ w.e.), and with respect to a steady state glacier with present-day length (3260 meter)."*

**Line-by-line comments**

> **>>** Line 14: Better to say already in the abstract which future climate data you used.

Agreed. This was added:

*"Future projections using CMIP5 temperature and precipitation data exhibit…"*

**>>** Line 24-25: This sentence needs some references.

We have included some references to these sentences:

*"… changing climate (e.g. Shannon et al., 2019; Zekollari et al., 2019; Hock et al., 2019)."*

**>>** Line 40: Change '.,' to ','.

Done.

**>>** Line 44-45: Are you referring to the whole Caucasian region?

Yes, the reference is updated, and it is stated explicitly that this encompasses the whole region:

*"…debris coverage has expanded at a rate of ca. +0.22 % yr⁻¹ between 1986 and 2014 when the entire Caucasus region is considered (Tielidze et al. 2020)."*

**>>** Line 65: you cannot use one glacier as representative for a whole area (Huss, 2008). However, at lines 78-79 it becomes clearer what you meant. So, please reformulate.

To avoid confusion, it is clarified in the updated manuscript:

*"… the behavior of the Djankuat Glacier as a WGMS reference glacier for the Caucasus …"*

**>>** Line 89: Give a number for 'higher elevations'

It was added to the updated text:

*"… higher elevations (> 3600 meter) and the…"*

**>>** Line 101: (i) it is not clear which mean annual temperature you are referring to (mean temperature of the 1981-2010 period?) (ii) it is not clear what 'here' is referring to.

(i) We refer to average annual temperatures. (ii) The word 'here' refers to both the Terskol and Mestia meteo stations. This is now clarified in the updated manuscript:

*"The average annual mean temperatures in Terskol and Mestia are 2.8°C and 6.0°C respectively during the 1981−2010 reference period. For the summer half-year from April to September (AMJJAS), the corresponding mean temperatures are 8.7°C and 12.0°C."*

Likewise, for precipitation:

*"At Terskol and Mestia, the average total precipitation amounts equal 1001.1 and 1035.1 mm yr⁻¹ w.e. respectively for the 1981−2010 climate. During the accumulation season (October to March, ONDJFM), the corresponding precipitation values are 418.4 and 490.0 mm yr⁻¹ w.e. respectively."*

**>>** Line 106: (i) you mentioned two places and then you say that you used only one automatic weather station (AWS). Did you used the same AWS in the two places? (ii) 'was installed' –> can you add from when to when? This information is especially important if you used only one AWS for two places.

(i) We used data from 2 AWSs (one in the Adylsu Valley near the LIA extent of the glacier, AWS 1, and one in the glacier ablation zone, AWS 2). We used data from AWS 1 for precipitation comparisons between Terskol and the Adylsu Valley, and data from AWS 2 to derive transmissivity, temperature lapse rates, albedo, and shortwave, longwave and turbulent fluxes. We now specifically refer to each AWS when data are discussed in the text. For Line 147:

> *"Hence, a direct comparison of measured air temperatures between AWS 2 on Djankuat and the Terskol weather station was found…"*

For Line 150:

> *"In this study, a value for $f_e$ of 1.5 between Terskol and the Adylsu Valley was found after a comparison of precipitation amounts from AWS 1 in the glacier valley."*

For Line 179:

> *"Measurements of the incoming solar radiation from the AWS 2 were used to derive atmospheric transmissivity…"*

For Line 185:

> *"The ice albedo $\alpha_{ice}$ can, according to raw data from the AWS 2, vary between 0.15 and 0.40 depending…"*

And Line 191:

> *"Here, these fluxes, as derived from AWS 2, are added up and plotted analyzed against air temperature following the method…"*

The location of AWS 1 and 2 were added to Figure 1 of the original manuscript. (ii) These AWSs were only operational during the summer months (June to September) between 2007 and 2017. This was clarified manuscript as follows:

> *"In 2007, two automatic weather stations (AWS) were additionally installed, one in the Adylsu Valley at ca. 2640 m elevation (AWS 1 in Fig. 1) and one in the ablation zone of the glacier at ca. 2960 m on a sparsely debris-covered ice surface (AWS 2 in Fig. 1). During the summer seasons (June to September) of 2007−2017, a wide range of additional meteorological variables have therefore been acquired by AWS 1 and 2 (air temperature, dew point temperature, incoming and outgoing shortwave/longwave radiation, relative humidity, wind speed and direction, air pressure and for AWS 1 also precipitation amounts) (Rets et al., 2019). The AWSs did not operate outside the JJAS period."*

**>>** Line 114: Maybe add 'glacier' before 'top', so that it becomes 100% clear.

Done.

**>>** Line 117-119: Sorry, I cannot follow this sentence. Can you maybe reformulate it?

The main point here is that we want to make our 1D flow line representative for the 3D glacier to not misestimate the rate of glacier shrinkage. To make this clearer, it was reformulated as:

> *"To avoid creating a bias in the rate of glacier evolution, the representativeness of the glacier cross-section along the flow line was further determined..."*

**>>** Line 121 (eq1): The way Eq. 1 is cast looks somewhat unusual to me. Can you add a reference where the derivation can be looked up? Or add the derivation in the manuscript?

The continuity equation in this form was discussed and derived by Oerlemans (2001).

**>>** Line 134: Spell out 'FTCS'.

Done and added to the text:

> *"… FTCS (forward in time, centered in space) numerical scheme…".*

> **>>** Line 141: Remove 'specific', since it is the glacier wide balance here.

The terminology in the literature is not consistent in this aspect. We choose to leave it like this.

> **>>** Line 145: Is there one value of ACC for the whole glacier, is it evaluated along the central flowline, or is there some sort of spatial grid playing a role?

It was derived for every point along the flow line. This was clarified in the text:

> *"Accumulation for each point along the flow line is only dependent…"*

> **>>** Line 149: (Oct-Mar) add the day, or whether the beginning/end of the month are meant.

Done. The text was updated to:

> *"Due to lack of AWS data outside of the JJAS period, a temperature lapse rate of -0.0049°C m⁻¹ was used for the winter half-year (1 Oct – 31 Mar), in accordance..."*

> **>>** Line 154: Add link to table 1 already after 'gamma_p'

Done.

> **>>** Line 144-159: Not super clear to me, especially how exactly all these factors are derived.

Precipitation between the Adylsu Valley and Terskol was scaled using a factor $f_e$. For the precipitation gradient, we did not have reliable data to extrapolate the precipitation from the Adylsu Valley over the entire glacier. We therefore used the accumulation profile to tune the precipitation gradient. The derivation of the parameters has now been described more extensively in the revised manuscript. Firstly, it is clarified where the factor $f_e$ comes from (See also comment Line 106):

> *"In this study, a value for $f_e$ of 1.5 between Terskol and the Adylsu Valley was found after a comparison of precipitation amounts with AWS 1 in the glacier valley."*

Secondly, it was explicitly stated that the precipitation gradient $\gamma_P$ is used as a tuning parameter:

> *"… by making use of a vertical precipitation gradient $\gamma_P$, of which the latter is used as a tuning parameter due to a lack of data (see Sect. 3.1)."*

Thirdly, with respect to $f_{red}$, for which a description was already pesent in the text, some additional info was included into the revised manuscript as follows:

> *"Here, a topographic characteristic is used to parameterize snow addition or removal from the glacier surface. It was quantified by dividing the linear accumulation profile (without the redistribution factor) with the observed profile and correlating these anomalies to the laterally averaged surface slope $s$ along the flow line (e.g. Huss et al., 2009). As such, a polynomial fit was found. For slopes steeper than the threshold, removal of snow can hence occur, and is assumed to be proportional to the surface slope itself."*

> **>>** Line 167: And alpha? Add that alpha is the albedo.

Done.

> **>>** Line 180: About which 'tilt' are you speaking? Is the AWS station tilted?

The tilt of surface slope is not relevant to measure incoming solar radiation. This was adjusted:

> *"To derive atmospheric transmissivity, measurements of the incoming solar radiation from AWS 2 were used."*

> **>>** Line 189: 'more or less' - please use a synonym.

This is changed to:

> *"… are approximately equally important…"*

> **>>** Line 189: 'Table 1' - It took me quite a lot to find values that you were referring to. Can't it simply be added to the text?

We opted to put less values in the text to not oversaturate the text with numbers. Therefore, we synthesized most values in one single table.

> **>>** Line 192: 'plotted' – Is this the correct word? With 'plotted' I expect a Figure…

To avoid confusion, the choice of words was adjusted. In the text, this was modified as follows:

> *"… fluxes, as derived from AWS 2, are added up and analyzed against air temperature…"*

> **>>** Line 200-205: Is the implicit assumption that C_debris is homogeneous within the entire glacier body? Since that's unlikely to be true, the assumption should at least be discussed.

This assumption had to be made due to lack of abundant information for this parameter. A short section is added to the revised manuscript to discuss the assumption:

> *"Here, a constant value for $C_{debris}$ in space and time is assumed. The emphasis of this work is to investigate the effect of supraglacial debris on melt patterns and glacier geometry. Encompassing englacial debris pathways or the spatial distribution of englacial debris concentration would add more detail than warranted by the lack of reliable data."*

In this regard, another change was made in Sect. 7 of the original manuscript. Here, we constrained the up-glacier position of the debris input locations $x_{debris}$ to a maximum position of $x_{debris} = x_{ELA}$, where ELA is the equilibrium line altitude. This was addressed as (Line 454):

> *"… is initiated from $x_{debris} = x_{ELA}$, at $t_{debris}$ = 2035 with a magnitude of $F_{debris}^{input}$ = 1.5 m yr⁻¹. For $x_{ELA}$, we calculated the average position of the ELA during a window of ±15 years surrounding $t_{debris}$ in the 'no additional debris scenario' (Sect. 6.1), which hence varies for each climatic scenario. We therefore choose to not initiate debris fluxes from positions above the ELA, due to the neglect of englacial pathways in our debris model (see Sect. 2.5)."*

> *"… the debris input location $x_{debris}$ was changed to 80%, 60% and 40% of the distance between $x_{ELA}$ and $x_L$ (further downstream), …"*

> **>>** Line 208: 'at 1680 m from the highest point' where is this point? Maybe show in Fig.1.

The reader can locate this point on the map by identifying the margin of the up-glacier debris extent in Figure 1. The following text was added to the manuscript:

> *"… 1680 m from the highest point (just below the ELA, at 88% of the distance between the terminus $x_L$ and the ELA $x_{ELA}$), since it is…"*

> **>>** Line 209-211: the choice of stopping the debris input flux at a given glacier width sounds rather arbitrary. Also the fact that the debris input location x_debris is fixed in time (and not moving) causes some doubts. Both points seem to merit some discussion.

We link connectivity issues between the topographic debris source and the main glacier body to this assumption. Hence, by that time, the glacier has shrunk too much to ensure that debris fluxes reach the glacier surface. This was added accordingly to the text (Line 212):

> *"Connectivity issues between the topographic source and the main glacier are forwarded as the main reason to justify this modification of the Anderson and Anderson (2016) model. Hence,*

> *by that time, the glacier shrunk too much to ensure that debris fluxes could still reach its surface."*

With regards to the debris input location, we attribute a lack of direct observations regarding past or future (static or moving) topographic debris sources to this assumption. However, an archived (rather unclear) satellite image of the Djankuat glacier in Pasthukov (2011) seems to point out that there has been only minor up-glacier migration of debris on the main glacier body since the 1970s. This led us to believe that our assumption of a static debris source could indirectly be justified. It was added the following discussion (Line 209):

> *"It was chosen to keep the debris input location at a fixed position due to a lack of direct observations regarding past or future (static or moving) topographic debris sources. However, a comparison of present-day satellite imagery with those from the 1970s (Pasthukov, 2011) seems to point out that the debris patches exhibit only minor up-glacier migration of debris on the main glacier tributary and the debris-covered orographically left part of the snout (when seen from the downstream direction), hence indirectly justifying the assumption."*

**>>** Line 215 (eq 13): The variable 't_debris' is not introduced.

It is now indicated explicitly what the term means (Line 208 in the original manuscript):

> *"Hence, $t_{debris}$ is the time at which the topographic debris source firstly starts to release its mass flux towards the glacier surface."*

**>>** Line 225: Can you give some more details about the relationship which was found?

This relationship can be seen in Eq. 15 and Fig. 5d of the paper. It incorporates an exponential decay of the fractional area along the flow line. We added this to the text:

> *"…parameterized based upon the distance from the terminus $D_T$, for which an exponential relationship was found from observations…"*

**>>** Line 228: How is the debris-area growth factor G_A 'updated yearly'? One should be pointed at eq. 17 at this stage.

Done. We added a reference to eq. 17: *"(see Eq. 17 in Sect. 3.2)."*

**>>** Line 234: you took into account the melting reduction effect of debris, but what about the melting enhancement effect of thin debris (e.g. Østream, 1959)? Add discussion about this.

With respect to the inclusion of the melt-enhancing effect for thin debris, studies performed on Djankuat Glacier point to a low value of the critical thickness (0.03 m by Lambrecht et al., 2011 and 0.07 m by Bozhinskiy et al., 1986). The areal fraction of debris cover on the Djankuat Glacier that holds such thin thickness values is very small, so we believe that the ablation enhancement effect of debris plays a very minor role on Djankuat Glacier. Therefore, the inclusion of this factor was not included in the parameterization. The following section was added to the updated manuscript for justification (Line 224):

> *"The melt enhancement that may occur for a very thin debris cover was not implemented. Values in the literature of the critical debris thickness for the Djankuat Glacier vary from 0.03 m (Lambrecht et al., 2011) to 0.07 m (Bozhinskiy et al., 1986). The areal fraction of Djankuat Glacier that holds these thin thickness values is very small (Popovnin et al., 2015) and are therefore not believed to have a significant influence on the ablation of Djankuat Glacier."*

**>>** Line 247, 251, 254: 'this time period' – maybe re-state the time period. Use same unit.

We changed *"this time period"* to *"the 1967/68−2006/07 period"*. Consistency in the units was achieved by changing them to *"m yr⁻¹ w.e. m⁻¹"*.

>> Line 258: What's the meaning of 'between 0.18 and +/- 0.6 m'.

The ± means *"approximately"*, and the text was adjusted accordingly'.

>> Line 261: 'second a' to 'a second'

Done.

>> Line 267: Can you give numbers about the 'snow redistribution by wind/avalanche'?

In the upper part (> 3600 meter), the local mass balance of the glacier is reduced by ca. 76%. The following was added to the text (Line 94):

*"Moreover, the mass balance profile in these upper areas is significantly distorted (by ca. -76%) by snow redistribution processes (Pastukhov, 2011)."*

>> Line 270-273: Did you validate the model with the same data which were used also to calibrate the model? If not, specify which data you used. If yes, isn't there a different, independent dataset which can be used for model validation?

There are very few or no independent data to validate the model results. In the revised manuscript this was acknowledged by reformulating the text in several places, e.g. Line 308 (Sect. 3.3):

*"However, as with the mass balance and debris cover model, there are no, or only few, independent data to validate our model results with a sufficient degree of certainty."*

For Line 394-395, Sect. 5.2, the word 'validation' was removed:

*"It can thus be stated that the calibrated mass balance model performs well when forced with the observed Terskol climatic data, and that credibility can be assigned to the dynamic calibration procedure."*

The same was done for the statement in the conclusion (Line 495, Sect. 8):

*"… no artificial mass balance perturbations were needed, ensuring proper model calibration and credibility."*

>> Line 280: I don't understand what t_debris exactly is (cf. l. 215).

See comment Line 215 above.

>> Line 297: 'the bed was slightly adjusted' – how? Can you give some more details?

This was clarified in the text by adding the following:

*"Additionally, the bed width for the assumed trapezoidal-shaped cross section was slightly adjusted to ensure that the parameterization fits the observed area-elevation distribution for a total surface area of 2.688 km²."*

>> Line 326: Is the volume change a yearly volume change? If yes correct the unit.

Yes, *"annual"* was added.

>> Line 331: Please correct the unit/make it consistent with the rest of the manuscript.

Not sure what is unclear here.

>> Line 332: Can you add a reference or details about the 'e-folding length response time'?

The following was added for clarification:

*"The e-folding length response time (i.e. the time needed to achieve $(1 - e^{-1})$ or ~63% of the total length change) of Djankuat is in the order of …"*

RC 1 – Loris Compagno (rebuttal by Verhaegen et al. – tc-2019-312)

>> Line 334-335: Well, I imagine that these numbers depend a lot on the topography (see general comments)?

See general comment 3.

>> Line 330-338: the sensitivity experiments need some more details on how they are done (see general comments).

See general comment 3.

>> Line 356-357: 'an acceptable accuracy' – please give a number.

The integrated mass balance of the glacier in steady state exhibits a value of $0 \pm 0.006$ m yr$^{-1}$ w.e., which was added to the text in the following way:

*"…integrated surface mass balance over the entire glacier approaches zero to within an acceptable accuracy of 0.006 m yr$^{-1}$ w.e. and by…"*

>> Line 381-387: In my opinion, all these sentences can be reduced into one or two.

The subsection was shortened to:

*"Especially during the last few decades, an accelerated warming trend has occurred, as the latest 10-year climatic interval exhibits a mean annual temperature anomaly of +0.5°C compared to the 1981−2010 mean. This makes it the warmest period in the reconstructed time series. For temperature, a clear sequence of colder and warmer intervals can be seen. Changes in precipitation show a sequence of drier and wetter periods (Fig. 8)."*

>> Line 390-395: How is this mass balance perturbation obtained (see main comments)?

See general comment 2.

>> Line 411-413, 413-415, 417: Can you give some more details about these data? Did you use global circulation models (GCMs) or regional climate models (RCMs)? Which model did you used (all GCMs and RCMs have specific names, realizations, institutions…). Why did you use a linear temperature and precipitation evolution? Why not using the transient evolutions of the climate models? Moreover: did you applied a de-biasing approach between past dataset and the future climate projections? (see main comments). '+7.1°C' compared to when? Is this number a temperature difference between two periods or a temperature mean? If it is a temperature mean, please report also the value of the first period, or the difference.

See general comment 1.

>> Line 445-456: Are these values arbitrary? That seems fine but if they are, better add a sentence explaining why these values are taken and from where.

These values are indeed arbitrary. It was added after Line 459 of the original manuscript:

*"It must be noted that the values of these parameters are arbitrary, as the exact location, time and magnitude of future debris sources cannot be predicted. By assessing multiple possible values for each of these parameters, we encompass various potential future scenarios in order to account for the high uncertainty regarding these parameters."*

>> Line 503: '-80%' – of area? Of volume? Or of length?

The following was changed (with a slight deviation due to the new future climatic datasets):

*"… most drastically (ca. -93 % of its current surface area) under the RCP 8.5 scenario…"*

**Comments to figures and tables**

>> Fig. 1: The scale bar is bizarre. Why not 0, 0.25, 0.5, 1 km instead of 0.5, 0.25, 0, 0.5 km?

We don't think the scale bar in this form is confusing.

>> Fig. 2: Would be nice to have a little map showing where these other glaciers are, or at least their distance to Djankutan Glacier.

The distances and direction to Djankuat Glacier are added to the updated Figure 2. Balance years are changed to calendar years and all length changes are now relative to 1900 AD:

[Figure]

*"New Figure 2 in the updated manuscript. Historic length variations of the Djankuat Glacier compared to other glaciers in the Caucasus (Solomina et al., 2016; WGMS, 2018). Approximate distances and direction to the Djankuat Glacier are indicated."*

>> Fig. 4: (i) I may have missed it, but what is causing the modelled MB gradient to 'flip' at the highest elevations (>3550 m) in Fig. 4a? (ii) Also the units are different than in main text.

(i) This is due to snow redistribution due to avalanches/wind redistribution in the steep upper part of the glacier (Lines 154-159 in the original manuscript). See comment to Line 267 above.
(ii) All mass balance terms are expressed in m w.e. $yr^{-1}$.

>> Fig. 5: (i) Would be interesting to see longer-term evolution of these values as well. (ii) What's causing the sharp bend around 1985? Is it the switch to very negative MBs

(i) An additional figure with the further evolution of the debris layer was not added to the manuscript in order to not overload the paper. The effect of additional debris sources can however be inferred from Fig. 14 in the original paper. (ii) The bend around the 1980s is when the debris that had been deposited at $x_{debris}$ = 1680 meter since $t_{debris}$= 1958, has reached the glacier terminus due to advection.

>> Fig. 10: (i) add labels for temperature and precipitation, (ii) say what "w.r.t." is and (iii) say what the black line is.

Labels were added, "w.r.t" has been deleted and the data have been converted from balance years to calendar years:

[Figure]

*"New Figure 8 in the updated manuscript. Reconstructed and observed evolution of (a) mean annual temperature and (b) total precipitation for Terskol weather station, based upon proxy data (tree ring reconstructions) and measurements from nearby weather stations (Mestia, Pyatigorsk and Mineralnye Vody). The dashed horizontal line represents the 1981−2010 annual reference values (2.6 °C and 1001.1 mm w.e. yr⁻¹). We refer to the text and Table 2 for more details."*

**>>** Fig. 11: What is the black line?

This is the 15-yr moving average. This is now clarified in the figure caption.

**>>** Fig. 12: Can the volume be plotted as well? That seems an important quantity.

We have opted to not overload the picture with an additional volume projection. Rather we chose to merge total annual runoff volumes (old Fig. 13) into the figure of the future projections, in the light of future water resource management. The results of the new future projections are represented in the following updated Figure 11:

[Figure]

*"New Figure 11 in the updated paper. Modelled (a) glacier length, (b) glacier surface area, and (c) total annual runoff volume of the Djankuat Glacier for different RCP scenarios until 2100 AD. In (c), the thin lines represent annual values, while the thicker lines represent 15-yr moving average. The dashed vertical line denotes the present (i.e. 2017, the most recent year of glaciological observations)."*

>> Fig. 13: (i) unit in the y-axis (/year?), (ii) say in the caption what 'present' is for this study.

This figure has been deleted and incorporated in the updated Figure 11 (see comment above).

>> Fig. 14: Why are the impacts visible only after ca. 2050?

The following statement is added to the manuscript:

*"It is worth mentioning that the effects on glacier length are not immediate, as it takes some time for the debris to be advected to the terminus."*

>> General comment about figures: The 'YYYY/YY' format is distracting. Better use 'YYYY'

Done for all figures that require annual labels on the x-axis.

>> Table 1: Can you explain what the '-' means?

The – means that there is no constant value for this parameter. We have added this as:

*"Table 1. Variables, constants and their units. The – denotes a variable or a dimensionless quantity."*

**New references added:**

Huss, M. & Hock, R: A new model for global glacier change and sea-level rise. Frontiers in Earth Science 3, 2015.

Hock, R., Rasul, G., Adler, C., Cáceres, B., Gruber, S., Hirabayashi, Y., Jackson, M., Kääb, A., Kang, S., Kutuzov, S., Milner, Al., Molau, U., Morin, S., Orlove, B., and Steltzer,H.: High Mountain Areas. In: IPCC Special Report on the Ocean and Cryosphere in a Changing Climate [Pörtner, H.-O., Roberts, DC., Masson-Delmotte, V., Zhai, P., Tignor, M., Poloczanska, E., Mintenbeck, K., Alegría, A., Nicolai, M., Okem, A., Petzold, J., Rama, B., Weyer, N. M. (eds.)]. In press, 2019.

Shannon, S., Smith, R., Wiltshire, A., Payne, T., Huss, M., Betts, R., Caesar, J., Koutroulis, A., Jones, D., and Harrison, S.: Global glacier volume projections under high-end climate change scenarios, The Cryosphere, 13, 325–350, doi: https://doi.org/10.5194/tc-13-325-2019, 2019.

Taylor, K. E., Stouffer, R. J., and Meehl, G. A.: An overview of CMIP5 and the experiment design, Bull. Am. Meteorol. Soc., 93, 485–498, doi: 10.1175/BAMS-D-11-00094.1, 2012.

Zekollari, H., Huss, M., and Farinotti, D.: Modelling the future evolution of glaciers in the European Alps under the EURO-CORDEX RCM ensemble, The Cryosphere, 13, 1125–1146, doi: https://doi.org/10.5194/tc-13-1125-2019, 2019.

---

## Author Comment (AC2) · 10 Jul 2020

Thank you for your detailed and helpful comments and suggestions. In the text below, reviewer comments are indicated with colored background, our replies are in plain text and our changes to the manuscript are put in italic.

**Response to major comments**

**Major comment 1**

>> Evaluation of glacier change in terms of terminus position and glacier area: We know that debris-covered glaciers have a different response to climate warming based on remote sensing observations and numerical modelling, which shows that they lose the majority of mass by surface lowering rather than terminus recession. Therefore, the metrics that are useful for clean-ice glaciers are poor indicators of the behaviour of a debris-covered glacier. My main concern with the study is that the debris model is unnecessary given the characteristics of Djankaut Glacier (e.g. large areas of visible clean ice on the tongue, steep slope below the ELA, high velocities, large changes in length and area over several decades, short response time) and introduces a bias to the results. It would be valuable to demonstrate the difference between simulations with and without the debris-cover model to evaluate its impact on glacier change and if the observed change can be replicated without this additional calculation. It appears that the info is contained in Fig. 14, which shows future glacier evolution under different climatic forcings, but is not discussed in the text and the figure is difficult to interpret; it appears that the debris has no impact on glacier length change until the second half of the century.

Indeed, our experiments brought to light that an extensive debris cover on Djankuat Glacier is a more recent phenomenon, largely linked to glacier retreat exposing debris sources, however that was not made very explicit in the manuscript. On the other hand, debris cover becomes an important characteristic of the glacier in the future. In that sense, we disagree with the reviewer that the debris cover is unnecessary to study the future behavior of Djankuat Glacier. We have more explicitly addressed this issue by including the results of an additional experiment without debris cover. Both model runs with and without debris cover exhibit very similar results prior to the observational period. As shown in the new inset in Fig. 9 below, debris played only a minor role prior to ca. 1980 AD, with length differences of only 20 to 40 meter. By 2009/10 AD, however, the length difference between both runs is already modelled to be 160 meter. This is also evident from observations, where one can clearly see that the debris-free section of the snout has retreated faster than the debris-covered section. In the manuscript, the following additional explanation was therefore added (Line 409):

*A historic model run conducted with a 100 % clean-ice glacier, shown as an inset in Fig. 9a, revealed that debris played only a minor role prior to ca. 1980 AD, with length differences of only 20 to 40 meter. By 2009/10 AD, however, the modelled length difference between a debris-free and debris-covered glacier already increased to 160 meter."*

And (Line 451):

*"Despite present-day areas of visible clean ice on the tongue, a steep slope below the ELA, relatively high ice velocities, and a short response time, also observations show that the supraglacial debris cover on the Djankuat Glacier has significantly affected glacier geometry during the last several decades, as evident from the differential retreat of the snout (Fig. 1).*

[Figure]

*Updated Figure 9. Historic variations of (a) the modelled and observed glacier length of the Djankuat Glacier since 1752/53 AD until 2017 AD, (b) additional mass balance perturbations ΔB$_a$ and (c) reconstructed time series of the total annual mass balance B$_a$ of the Djankuat Glacier with changing geometry. Observed length variations are derived from lichenometric dating of moraines in the valley, historic documents, and/or field measurements and/or recent satellite imagery (Boyarsky, 1978; Zolotarev, 1998; Petrakov et al., 2012; WGMS, 2018). An additional model run for a 100% clean ice glacier was conducted is shown in the inset in panel a.*

We have furthermore expanded the discussion of Figure 14 to underline that supraglacial debris cover is of large importance for the future evolution of the glacier:

*The figure shows the impact of debris input location x$_{debris}$, the time of release of the debris source from the surrounding topography t$_{debris}$, and debris flux magnitude F$_{debris}$ (rows) on the future length extension of the Djankuat Glacier under different climatic scenarios (columns). The black lines indicate the scenario where no additional debris source is released in the future. The other lines are for experiments that include an additional future debris source from the surrounding topography for varying values of the earlier mentioned debris-related parameters. It is clear that the addition of an increasingly widespread debris cover dampens glacier retreat. It should be noted that the effects on glacier length are not immediate, as it takes some time for the debris to be advected to the terminus after its initiation at time t$_{debris}$.*

**Major comment 2**

**>>** Value of sub-debris melt calculation: In relation to my point above, I have two concerns about the debris-cover model; (a) the gradient of the exponential function used to scale sub-debris melt is steep using H*debris = 1.15 m (see review Fig. 1), and (b) the thickness of debris on the glacier is similar to the critical thickness observed on debris-covered glaciers elsewhere and therefore likely to both enhance and reduce ablation across the tongue. The glacier model accounts for the impact of supraglacial debris by reducing mass balance, a valid assumption

> beneath debris that is thicker than a critical thickness of about 0.1–0.2 m. An exponential function is used to reduce ablation with debris thickness. However, images of the present day glacier including Fig. 1 and data presented in the manuscript (Fig. 5a) illustrate that the debris thickness at the terminus is ~1.0 m in 2010 and was <0.5 m before 1990. As debris thickness decreases rapidly upglacier (Fig. 5c), is the same scaling is assumed then most of the debris layer was <0.25 m thick before 1990 and therefore close to the critical thickness. For such thin and discontinuous debris layers, there is likely to be little reduction in ablation due to insulation by the debris layer (the exponential function used here will only reduce sub-debris melt by <20% compared from the clean-ice value – see review Fig. 1) and instead an enhancement of ablation due to the reduction in albedo of debris-covered ice compared to clean-ice surfaces. The model does include an albedo term but does not use this to adjust for the impact of debris on ablation.

We understand the reviewers' concern related to the decay of the exponential curve for H*debris and acknowledge that the value found for Djankuat Glacier deviates somewhat compared to earlier research for other glaciers. As pointed out in Anderson and Anderson (2016) and Lambrecht et al. (2011), the value for H*debris depends, amongst other factors, on the thermal conductivity of the debris material, the debris cover porosity and is also influenced by the debris layer water saturation. Values for these factors seem somewhat out of range for the Djankuat Glacier and explain the deviating value of the H*debris parameter (Anderson and Anderson, 2016; Lambrecht et al., 2011; Bozhinskiy et al., 1986). The following section was added to the manuscript for clarification (Sect. 3.1, Line 262):

> *"A value of 1.15 meter was found for H*debris. The gradient of the exponential decay is somewhat out of range with respect to earlier studies for other glaciers (e.g. Anderson and Anderson, 2016). Explanations for this high value of H*debris can be found in the relatively high thermal conductivity of the granite-type debris cover on the glacier (2.8 W m$^{-1}$ °C$^{-1}$) and the high debris cover porosity (0.43 in the case of Djankuat Glacier, Bozhinskiy et al., 1986). Also the relatively low water saturation, as mentioned by Lambrecht et al. (2011), suggests that heat conduction towards the debris-ice interface seems to occur quite easily on the Djankuat Glacier."*

With respect to the inclusion of the melt-enhancing effect for thin debris, studies performed on Djankuat Glacier point to a lower value of the critical thickness than mentioned by the reviewer (0.03 m by Lambrecht et al., 2011 and 0.07 m by Bozhinskiy et al., 1986). The areal fraction of debris cover on the Djankuat Glacier that holds such thin thickness values is very small, so we believe that the ablation enhancement effect of thin debris plays a very minor role on Djankuat Glacier. Therefore, this factor was not included in the parameterization. The following section was added to the revised manuscript for justification (Line 224):

> *"The melt enhancement that may occur for a very thin debris cover was not implemented. Values in the literature of the critical debris thickness for the Djankuat Glacier vary from 0.03 m (Lambrecht et al., 2011) to 0.07 m (Bozhinskiy et al., 1986). The areal fraction of Djankuat Glacier that holds these thin thickness values is very small (Popovnin et al., 2015) and are therefore not believed to have a significant influence on the ablation of Djankuat Glacier."*

The following limitations were furthermore added for completion, after Line 228:

> *"The debris model also neglects other processes that may potentially play a role in the spatial and temporal distribution of debris, such as the formation and thickening of medial moraines, ice cliffs and surface ponds (Anderson and Anderson, 2016)."*

**Major comment 3**

> **>>** Evolution of the debris layer: As observed in Fig. 5, the debris layer has thickened by a factor of 2–3 over the last 20 years. Djankaut Glacier is steep, fast-flowing and thinly debris-

> covered over a section of its ablation area, and based on this geometry and the presence of large ice-marginal moraines it seems likely that during the LIA and subsequently, the glacier exported the majority of its debris to its margins rather than developing a supraglacial layer. Therefore, the assumption in the spin up simulation that the glacier is debris covered (Section 5.1) may not hold. However, from Fig. 14 it appears that the debris layer has no impact on glacier change until about 1970 CE.

We agree that the increasingly widespread supraglacial debris cover on Djankuat Glacier is a more recent phenomenon, largely related to exposure of debris sources due to glacier retreat and climate warming. We furthermore refer to the new Fig. 9a in general comment 1 to demonstrate that the debris cover only became important during the last several decades, and has had little influence prior to ca. 1980 AD. However, there is also an indirect evidence for the presence of at least some supraglacial debris in the historic period, shown by e.g. the presence of end moraines in the valley (Fig. 1), and a photograph taken around 1930 AD that shows some debris patches on the snout (Aleynikov et al., 2002). It is furthermore unclear to us how we could have initialized the glacier model at the LIA without debris cover. The following section was therefore added to the manuscript to discuss this issue (in Sect. 5.1):

> *"As can be deduced from the large lateral moraines in the Adylsu Valley (Fig. 1) and fast-flowing nature of the paleo-glacier tongue in the valley (up to 100 m yr$^{-1}$ around 1752 AD, Fig. 6d), Djankuat Glacier used to export most of its debris to the margins rapidly in the historic period, rather than developing a supraglacial debris cover. Furthermore, debris sources from surrounding topography were likely less widespread in the historic period because the slopes were covered by the glacier itself and were more stable in a colder climate. For this reason, supraglacial debris is believed to have been much less widespread prior to the observational period of 1967/68 AD, implying that the glacier was not very much influenced by debris cover in the historic period. However, there is also indirect evidence for at least some supraglacial debris in the historic period from the presence of end moraines in the valley (Fig. 1) and a photograph taken around 1930 showing some debris patches on the snout (Aleynikov et al., 2002). It would be furthermore unrealistic to only introduce a debris cover in the model once the model approaches the start of the observations. This would contradict the presence of moraines and the observation that there already was an expanding debris cover during the first data collection in 1967/68 AD (Popovnin et al., 2015). Because there is no direct evidence for the origin of the debris cover, it was chosen to include melt-out processes in the model from the initialization onwards."*

As a minor point, Fig. 14 (now Fig. 12) only showed results after 2009, so we are a bit puzzled how reviewer 2 came to the conclusion that the debris layer has no impact until about 1970 CE.

**Major comment 4**

> **>>** Lack of discussion: The manuscript organisation is somewhat unconventional. After the Introduction, Methods and Model Description, there are four Results sections (not named as such) followed by the Conclusions. There is very limited discussion of the context of results and their interpretation, and no dedicated section for this.

We organized the manuscript in such a way that discussion items are merged into the result sections, so that all information related to one specific subject appears sequentially in a chronological, continuous text. This way of structuring was preferred, rather than jumping from one section to another. However, also in response to the other reviewers, the discussion was expanded in several places. This included additional discussion on model validation versus model calibration, justification of assumptions in the debris cover model, and the effect of debris on future glacier evolution.

**Response to minor comments**

> **>>** Line 13: "retreat" see major comment 1 about terminus recession versus surface lowering.

To elaborate more on the thinning out of debris-covered glaciers, we added (Line 51):

*"If a thick supraglacial debris cover is present over a large portion of a glacier's ablation zone, surface melting and terminal retreat can be drastically suppressed, even under a warming climate (e.g. Scherler et al., 2011; Benn et al., 2012). In such cases, debris-covered glaciers are shown to lose mass by lowering the surface in their ablation zone (downwasting), rather than by terminus retreat (e.g. Hambrey et al., 2008; Rowan et al., 2015)."*

> **>>** Line 15-16: The change in glacier length and area stated here are not meaningful unless the initial length and area are also given, or these are stated as % change.

Done. This was rectified in the text:

*"… have decreased by 1.4 km (- 29.5 %) and 1.6 km² (-35.2 %) respectively…"*

> **>>** Line 24-25: Vague statement.

We have included some references to these sentences:

*"… changing climate (e.g. Shannon et al., 2019; Zekollari et al., 2019; Hock et al., 2019)."*

> **>>** Line 39: Use of "significantly" should be reserved to its precise statistical meaning, whereas here it is used for emphasis and could be replaced with "dramatically" or in this sentence the meaning would be the same if this word was removed.

The word 'significantly' was removed and replaced by 'drastically'.

> **>>** Line 45-47: What is the glacierised area and debris-covered area in the Caucasus in km2? This is needed to indicate the context suggested in this statement.

The total glaciated area is stated in Line 30 (691.5 ± 29.0 km² in 1986, 590.0 ± 25.8 km² in 2014). The manuscript mentioned on Line 46 that 26.2% of that glacierized area is debris covered, referring to Scherler et al. (2018). Hence, the debris-covered area is ca. 155 ± 6.7 km² for present-day conditions. This number is now added:

*"…be 26.2 % at for present-day conditions (ca. 155 ± 6.7 km²), hence enabling…"*

> **>>** Line 53: Citations to previous modelling studies of debris-covered glaciers. Please note that Rowan et al. (2015) did not use a simple parameterisation of the impact of debris on mass balance as stated here, but instead made a dynamic simulation of the feedbacks between ice flow, debris transport and mass balance using a higher-order ice flow model. The statement ending in line 64 is therefore incorrect, as previous studies have taken this approach. A citation to Wirbel et al. (2017) should also be included.

This has been rectified in the text, thanks for pointing this out. The sentence was changed to:

*"The pronounced effect of debris should not be ignored in numerical models to determine the future evolution of mountain glaciers, yet only few studies have included this complex process in time-dependent models (e.g. Jouvet et al., 2011; Rowan et al., 2015; Huss and Fischer, 2016; Kienholz et al., 2017; Rezepkin and Popovnin, 2018; Wirbel et al., 2018)."*

> **>>** Line 73: State glacier area here.

Done.

>> Line 96-98: Use metres for debris thickness values here to be consistent with the rest of the text.

Done.

>> Line 101: "Mean annual air temperature", and "+" is not needed before the values.

Changed.

>> Line 110: Explain what you mean by "1.5D" or stick with "1D" to indicate a flow line calculation. L112. Do you mean 2D rather than "3D", i.e. a matrix calculation?

The model only uses ice and debris flow in 1 dimension, namely along the x-axis. However, the remaining glacier area was also implicitly taken into account by using the width in the continuity equation. To avoid confusion, it was changed to *"numerical flow line model"*.

>> Line 224: Give value for H*debris, from Table 1, the value used after tuning was 1.15 m, which results in the steep curve mentioned in Major Comment 1. Also it is not clear as written here how this model compared to that presented in Anderson and Anderson (2016) as mentioned in the Introduction, which used a hyperbolic rather than exponential function to scale sub-debris melt; h*/(h*+hdebris) their Eq. 3 with h* of 0.065 m.

See major comment 2.

>> Line 258, 260: Unclear as written. What is the meaning of "±" before the values given for H*debris? Do these values range from –0.6 to 0.6 m?

The ± means *"approximately"*, and the text was adjusted accordingly'.

>> Line 259: One of the key references for a previous application of this model to Djankaut Glacier is Rybak et al. (2018), which is cited to justify parameter choices and to give detail about the model. However, this document is difficult to locate and appears to only be available in Russian. I was not able to use this reference to collect information about the model. At Line 259 the citation here is incorrect, as "Rybak (2018)" is not in the reference list.

We acknowledge that both Russian papers are hard to find and not easy to understand, and have therefore decided to remove these from the manuscript.

>> Line 363: All the models have different time steps; 3-hourly for the mass balance model, ~4 hourly for the ice flow model and ~4 days for the debris transport model. How are the integrated, and what impact do these time steps have on the result when the response time is ~30 years?

The time steps for the ice flow and debris models were chosen for reasons of numerical stability to satisfy the CFL criterion for diffusion and advection problems. The timestep of 3 hours for the mass balance model is required to capture the daily cycle and because the weather data were not available at shorter intervals. The mass balance is calculated for a full balance year, changing year per year. The choice of these time steps has a negligible impact on the results given the length response time of ca. 31 years.

>> Line 455-459: What evidence is there for the choice of debris input parameters?

These values represent a range of possible future scenarios informed by the past, as the location, release and magnitude of future debris sources can of course not be predicted. We added the following text after Line 459 for clarification:

*"It must be noted that the values for these parameters represent a range of possible future scenarios, as the exact location, time and magnitude of future debris sources cannot be predicted."*

**>>** Line 490: Incorrect statement, see comment on line 53 above.

Agreed. Changed to:

*"… not yet integrated in numerical flow line models."*

**>>** Line 508-518: Here and elsewhere, although the written text is generally clear and free of typographic errors, the writing style is rather vague and qualitative, using large lists of variables/controls without indicating their importance, and the meaning can be difficult to follow. The manuscript would benefit from editing to enable clearer, more precise statements to present the study and its results.

Noted.

**>>** Model code: The code and data used are described as available on request from the author. I believe the Cryosphere now requires these to be open access in a repository.

To comply with TC's data policy, we now make the model code publicly available via GitHub/Zenodo. The model code that served for this research can be found and downloaded from: https://github.com/yoniv1/Djankuat_glacier_model. The code placed here is a 1D coupled ice flow-debris cover model. It uses bedrock geometry together with a parameterized mass balance profile to calculate the ice thickness evolution on a grid with spatial resolution dx for the Djankuat Glacier, and also takes into account an evolving supraglacial debris cover until a steady state situation has been reached. Our code availability statement now reads:

*"Code availability. Code availability. The coupled ice flow-supraglacial debris cover model for the Djankuat Glacier used in this research was written in MATLAB_R2019a. It can be downloaded from the GitHub repository at: https://github.com/yoniv1/Djankuat_glacier_model, doi: https://doi.org/10.5281/zenodo.3934612."*

**Newly added references**

Aleynikov, A. A., Zolotaryov, Ye. A., Voytkovskiy, K. F., and Popovnin, V.V.: Indirect Estimation of the Djankuat Glacier Volume Based on Surface Topography, Hydrology Research, 33 (1), 95–110, doi: 10.2166/nh.2002.0006, 2002.

Benn, D. I., Bolch, T., Hands, K., Gulley, J., Luckman, A., Nicholson, L. I., Quincey, D., Thompson, S., Toumi, R., and Wiseman, S.: Response of debris-covered glaciers in the Mount Everest region to recent warming, and implications for outburst flood hazards, Earth Sci. Rev., 114, 156–174, doi:10.1016/j.earscirev.2012.03.008, 2012.

Hambrey, M., Quincey, D., Glasser, N. F., Reynolds, J. M., Richardson, S. J., and Clemmens, S.: Sedimentological, geomorphological and dynamic context of debris-mantled glaciers, Mount Everest (Sagarmatha) region, Nepal, Quaternary Sci. Rev., 27, 2341–2360, 2008.

Hock, R., Rasul, G., Adler, C., Cáceres, B., Gruber, S., Hirabayashi, Y., Jackson, M., Kääb, A., Kang, S., Kutuzov, S., Milner, Al., Molau, U., Morin, S., Orlove, B., and Steltzer,H.: High Mountain Areas. In: IPCC Special Report on the Ocean and Cryosphere in a Changing Climate [Pörtner, H.-O., Roberts, DC., Masson-Delmotte, V., Zhai, P., Tignor, M., Poloczanska, E., Mintenbeck, K., Alegría, A., Nicolai, M., Okem, A., Petzold, J., Rama, B., Weyer, N. M. (eds.)]. In press, 2019.

Scherler, D., Bookhagen, B., and Strecker, M. R.: Spatially variable response of Himalayan glaciers to climate change affected by debris cover, Nat. Geosci., 4, 156–159, 2011.

Shannon, S., Smith, R., Wiltshire, A., Payne, T., Huss, M., Betts, R., Caesar, J., Koutroulis, A., Jones, D., and Harrison, S.: Global glacier volume projections under high-end climate change scenarios, The Cryosphere, 13, 325–350, doi: https://doi.org/10.5194/tc-13-325-2019, 2019.

Wirbel, A., Jarosch, A. H. and Nicholson, L.: Modelling debris transport within glaciers by advection in a full-Stokes ice flow model, The Cryosphere, 12, 189-204, https://doi.org/10.5194/tc-12-189-2018, 2018.

Zekollari, H., Huss, M., and Farinotti, D.: Modelling the future evolution of glaciers in the European Alps under the EURO-CORDEX RCM ensemble, The Cryosphere, 13, 1125–1146, doi: https://doi.org/10.5194/tc-13-1125-2019, 2019.

---

## Author Comment (AC3) · 10 Jul 2020

Thank you for your detailed and helpful comments and suggestions. In the text below, reviewer comments are indicated with colored background, our replies are in plain text and our changes to the manuscript are put in italic.

**Response to general comments**

**General comment 1**

>> Model validation: at several places in the manuscript, the authors say that the model was validated, for example: "It can thus be stated that the model performs well and underwent a successful validation to within acceptable accuracy". I argue that a model is validated when its capacity to reproduce the "unseen" is assessed (past and future evolution, or unobserved variables). A model is useful when model predictions are associated with an uncertainty estimate. As it is now, the model has a very large number of free parameters which are calibrated to match observations almost perfectly. Per design, the study does not allow validation with independent or out-of-sample data (e.g. cross-validation). I don't think that it is possible to change this aspect of the study at this stage, but I would like to see the problem of model uncertainty and over-calibration discussed in the manuscript, and the statement that the model has been successfully "validated" should be changed to "calibrated". I think that the consequences of parameter equifinality are most likely to be seen in the sensitivity experiments of the debris cover parameterization and the future projections.

It is correct that there are very few or no independent data to validate the model results and we agree that 'validation' is not the appropriate choice of word in this regard. In the revised manuscript this was acknowledged by reformulating the text in several places, basically substituting 'validation' by 'calibration' (e.g. Line 308, Sect. 3.3):

*"However, as with the mass balance and debris cover model, there are no, or only few, independent data to validate our model results with a sufficient degree of certainty."*

For Line 394 (Sect. 5.2), we removed the word 'validated':

*"It can thus be stated that the calibrated mass balance model performs well when forced with the observed Terskol climatic data, and that credibility can be assigned to the dynamic calibration procedure."*

The same was done for the statement in the conclusion (Line 495, Sect. 8):

*"… no artificial mass balance perturbations were needed, ensuring proper model calibration and credibility."*

Furthermore, a small section was introduced related to over-calibration. Apart from a small areal fraction in the highest altitudinal zones (> 3600 meter), where data availability is limited and snow redistribution processes create complex patterns, we think that our calibration dataset is sufficiently long (39 years) to assume that the environmental conditions within the calibration window have some validity for past and future conditions (Sect. 3.1, Line 273):

*"The calibration dataset for the mass balance model is quite long (39 years from 1967/68 to 2006/07 AD), making it credible to assume that the parameters calibrated to this period have some validity for past and future conditions as well. Apart from the high-elevation areas (> 3600 meter), where data availability is limited and snow redistribution processes create complex conditions, it can be expected that the environmental setting within the calibration window also holds for periods prior to and after the observational period. However, it must be noted that the areal fraction of this high-altitudinal zone is limited (ca. 3% of the glacier area in 2009/10 AD)."*

**General comment 2**

>> Added value of the past simulations: the model is dynamically tuned to fit observed length changes, with a time varying bias parameter. I am aware that this has been done before (and will be done in the future), but I have to ask: in the end, what is the added value of such a simulation? What do we learn from it, that we didn't already know from length change observations alone? What are the implications of the dynamic parameterization for the future projections?

The dynamic calibration procedure is needed to account for imperfections in the model and the climate forcing datasets, which are generally larger for more distant time periods. Because the glacier is currently still responding to past changes of climate, geometry and dynamics, these imperfections would produce a current glacier state that deviates from the observed one, and this deviation would be carried forward in any projection. The positive aspect of our dynamic calibration is that mass balance corrections were only required for the period before 1967 AD, so that (keeping in mind the e-folding length response time of ca. 31 years) future projections have largely 'forgotten' the older artificial mass balance corrections. We further refer to general comment 2 of the Loris Compagno review (RC 1) and its responses. For clarification, the following was added to the text (Line 395):

*"Such a procedure is needed to counteract imperfections in the flow model, mass balance model and the climate forcing. The added value of this procedure is to ensure a current glacier state that matches the observed one, as the glacier is still responding to changes in past climate, geometry and dynamics."*

And:

*"It furthermore implies that future projections are no longer influenced by the corresponding artificial mass balance corrections, keeping in mind an e-folding length response time of ca. 31 years for the Djankuat Glacier."*

**General comment 3**

>> Debris cover parameterization: in my opinion, the true added value of this study lies in the coupling of a debris parameterization with the flowline model. I think it would add great value to the manuscript to extend the sensitivity analyses to the past glacier simulation as well (which, as it stands, is of very limited usefulness). How is the past glacier evolution changed by the inclusion of debris cover? In order not to make this paper even longer, I would suggest to remove Fig. 7 to 9, which are quite qualitative.

We believe that the extensive debris cover on Djankuat Glacier is a more recent phenomenon, largely linked to glacier retreat exposing debris sources. We therefore assume that the glacier was not very much influenced by debris cover prior to the observational period (1967/68 AD). In that sense, an experiment related to historic debris characteristics was carried out by executing a model run for the historic period, both with and without the debris parametrization. Both model runs exhibited very similar results prior to the observational period. As shown in the new inset in Fig. 9 below, debris played only a minor role prior to ca. 1980 AD, with length differences of only 20 to 40 meter. In this regard, the following was added to the paper (in Sect. 5.1):

*"As can be deduced from the large lateral moraines in the Adylsu Valley (Fig. 1) and fast-flowing nature of the paleo-glacier tongue in the valley (up to 100 m yr$^{-1}$ around 1752 AD, Fig. 6d), Djankuat Glacier used to export most of its debris to the margins rapidly in the historic period, rather than developing a supraglacial debris cover. Furthermore, debris sources from surrounding topography were likely less widespread in the historic period because the slopes were covered by the glacier itself and were more stable in a colder climate. For this reason, supraglacial debris*

*is believed to have been much less widespread prior to the observational period of 1967/68 AD, implying that the glacier was not very much influenced by debris cover in the historic period."*

And (Line 409):

*"A historic model run conducted with a 100 % clean-ice glacier, shown as an inset in Fig. 9a, revealed that debris played only a minor role prior to ca. 1980 AD, with length differences of only 20 to 40 meter. By 2009/10 AD, however, the modelled length difference between a debris-free and debris-covered glacier increased to 160 meter"*

[Figure]

*Updated Figure 9. Historic variations of (a) the modelled and observed glacier length of the Djankuat Glacier since 1752/53 AD until 2017 AD, (b) additional mass balance perturbations ΔB$_a$ used in the dynamic calibration procedure and (c) reconstructed time series of the total annual mass balance B$_a$ of the Djankuat Glacier with changing geometry. Observed length variations are derived from lichenometric dating of moraines in the valley, historic documents, field measurements and recent satellite imagery (Boyarsky, 1978; Zolotarev, 1998; Petrakov et al., 2012; WGMS, 2018). An additional model run for a 100% clean ice glacier is shown in the inset in panel a.*

To limit paper length, we removed Figs. 7 to 9 of the original paper and replaced it with a single all-encompassing figure regarding glacier sensitivity (see also our response to general comment 3 of reviewer RC 1).

**General comment 4**

>> Code and data availability: you write: "the refined debris cover implementation can be used for comparable glacier models in future research". I agree! But it would be considerably more useful if the code and data used in this study would be made freely available under a proper software license and in a public repository. Platforms like zenodo.org will preserve the version

of the model as it is at the time of this publication. And it will create a DOI to make it citable for future research. See TC's data policy: https://www.the-cryosphere.net/about/data_policy.html

You are right. To comply with TC's data policy, we now make the model code publicly available via GitHub/Zenodo. The model code that served for this research can be found and downloaded from: https://github.com/yoniv1/Djankuat_glacier_model. The code placed here is a 1D coupled ice flow-debris cover model. It uses bedrock geometry together with a parameterized mass balance profile to calculate the ice thickness evolution on a grid with spatial resolution dx for the Djankuat Glacier, and also takes into account an evolving supraglacial debris cover until a steady state situation has been reached. Our code availability statement now reads:

*"Code availability. The coupled ice flow-supraglacial debris cover model for the Djankuat Glacier used in this research was written in MATLAB_R2019a. It can be downloaded from the GitHub repository at: https://github.com/yoniv1/Djankuat_glacier_model, doi: https://doi.org/10.5281/zenodo.3934612."*

**Specific comments**

>> Abstract L10: I would prefer not to use the term "1.5D". I never understood what the "0.5" is referring to: the widths? The vertically integrated velocity? Should a 2D SIA model then be called a 2.5D model? I think that a "SIA flowline model" is explicit enough.

We have replaced the term '1.5D' in the original manuscript and now call the model 'flow line model' in the new text, so that its 1-dimensional nature can be derived explicitly from its name.

>> Figure 1: If possible, indicate the location of AWS Adylsu Valley

Done.

>> Figure 2: It is misleading to compare length changes like this, because they all have a different zero baseline. It would be better to plot them as relative length change since year X.

The figure was updated, where distances and direction to Djankuat Glacier are added. Balance years are changed to calendar years and all length changes are now relative to 1900 AD:

[Figure]

*"New Figure 2 in the updated manuscript. Historic length variations of the Djankuat Glacier compared to other glaciers in the Caucasus (Solomina et al., 2016; WGMS, 2018). Approximate distances and direction to the Djankuat Glacier are indicated."*

>> Figure 6a and chapter 5.1: Could you elaborate on why the 1752 steady-state glacier has a longer and thicker tongue while the ice thickness above 3000 m a.s.l. is more or less equal to the 2009 glacier?

This is a well-known characteristic of glacier retreat. Retreating glaciers thin their ablation areas with little effect above the equilibrium line, which is well reproduced by the model.

>> Figure 10 and 11c: I assume the black lines are rolling means. Please specify years.

These are 15-yr moving means. This was added to the figure caption.

>> Line 106: Please specify which data period and parameters are available at these stations.

We used data from 2 AWSs (one in the Adylsu Valley near the LIA extent of the glacier, AWS 1, and one in the glacier ablation zone, AWS 2). We used data from AWS 1 for precipitation comparisons between Terskol and the Adylsu Valley, and data from AWS 2 to derive transmissivity, temperature lapse rates, albedo, and shortwave, longwave and turbulent fluxes. These AWSs were only operational during the summer months (June to September) between 2007 and 2017. This was clarified manuscript as follows:

*"In 2007, two automatic weather stations (AWS) were additionally installed, one in the Adylsu Valley at ca. 2640 m elevation (AWS 1 in Fig. 1) and one in the ablation zone of the glacier at ca. 2960 m on a sparsely debris-covered ice surface (AWS 2 in Fig. 1). During the summer seasons (June to September) of 2007–2017, a wide range of additional meteorological variables have therefore been acquired by AWS 1 and 2 (air temperature, dew point temperature, incoming and outgoing shortwave/longwave radiation, relative humidity, wind speed and direction, air pressure and for AWS 1 also precipitation amounts) (Rets et al., 2019). The AWSs did not operate outside the JJAS period."*

The location of AWS 1 and 2 were added to Figure 1 of the original manuscript.

>> Line 134: The time step is fixed at 0.0005 years. Did any stability considerations or tests go into this choice?

The following was added to the updated manuscript (Line 135):

*"…with $\Delta t$ of 0.0005 years, as determined by the CFL-condition for diffusion problems".*

And Line 221:

*"…with $\Delta t$ = 0.01 years, in accordance with the CFL-condition for advection problems".*

>> Line 147, 179, 183: Which time period was used?

We used the data from the AWSs, which were only operational during the summer seasons of 2007-2017. We refer to comment Line 106 above.

>> Line 149: Was the winter temperature lapse rate solely chosen based on the reported ELA temperature by WGMS (2018) or was AWS data used as well?

The AWSs were not operational outside the June to September window. This was clarified:

*"Due to lack of AWS data outside of the June to September period, a temperature lapse rate of -0.0049°C m$^{-1}$ was used for the winter half-year (1 Oct – 31 Mar), in accordance..."*

>> Line 150, 179, 191: It is often not clear if data from the AWS Djankuat or the AWS Adylsu Valley is used.

We used data from AWS 1 for precipitation comparisons, and data from AWS 2 to derive transmissivity, temperature lapse rates, albedo, and shortwave, longwave and turbulent fluxes. We specifically referred to each respective AWS when data are discussed in the text. For Line 147:

*"Hence, a direct comparison of measured air temperatures between AWS 2 on Djankuat and the Terskol weather station was found…"*

For Line 150:

*"In this study, a value for $f_e$ of 1.5 between Terskol and the Adylsu Valley was found after a comparison of precipitation amounts from AWS 1 in the glacier valley."*

For Line 179:

*"Measurements of the incoming solar radiation from the AWS 2 were used to derive atmospheric transmissivity…"*

For Line 185:

*"The ice albedo $\alpha_{ice}$ can, according to raw data from the AWS 2, vary between 0.15 and 0.40 depending…"*

And Line 191:

*"Here, these fluxes, as derived from AWS 2, are added up and plotted analyzed against air temperature following the method…"*

**>>   Line 150, 250**: Is the precipitation scaled to match one of these AWS? If yes which one?

Precipitation between the Adylsu Valley and Terskol was scaled using a factor $f_e$, using AWS 1. For the precipitation gradient, we did not have reliable data to extrapolate the precipitation from the Adylsu Valley over the entire glacier. We therefore used the accumulation profile to tune the precipitation gradient. It is clarified where the factor $f_e$ comes from:

*"In this study, a value for $f_e$ of 1.5 between Terskol and the Adylsu Valley was found after a comparison of precipitation amounts with AWS 1 in the glacier valley."*

Secondly, it was explicitly stated that the precipitation gradient $\gamma_P$ is used as a tuning parameter:

*"… by making use of a vertical precipitation gradient $\gamma_P$, of which the latter is used as a tuning parameter due to a lack of data (see Sect. 3.1)."*

**>>   Line 162 (Eq. 7)**: It might be worth noting that this melt term is only one part of the total runoff of the mass-balance model and that the rest is derived in the next chapter.

Done:

*"It must be noted that the melt term M is only one part of the total runoff RO of the mass balance model (see Sect. 2.5)."*

**>>   Line 187**: Can you please specify how the fractional cloud cover is parametrized?

It was done using a linear relationship between the cloud cover and net longwave radiation. The following was added:

*"These were derived from an approximately linear relationship between the cloud cover and the net longwave radiation balance (Voloshina, 2002), of which the latter was measured by AWS 2 on the glacier surface."*

**>>   Line 192**: From that sentence I would expect a Figure similar to Figure A1 of Giesen and Oerlemans (2010).

To avoid confusion, the choice of words was adjusted. In the text, this was modified as follows:

*"… fluxes, as derived from AWS 2, are added up and analyzed against air temperature…"*

**>> Line 369**: "At first, data from the pre-observational period ..."

Done. We changed *"for"* to *"from"*.

**>> Line 370**: Terskol time period is already specified in Table 2.

The text *"(1977−2013 with a data gap between 1990−1997)"* was deleted.

**>> Line 373**: How was the available data repeated into the past? By just copying the entire time period? Shuffling of individual days/months/years? Were any sensitivity tests made in that regard?

As mentioned in the manuscript, the data sequence for Terskol over which measurements with a 3-hourly interval are available (1977−2013 with a data gap between 1990−1997) was repeated into the past and future in order to maintain intra-daily and intra-annual variability in the data. These 30-year sequences were copied / pasted until the entire time series had been covered. Afterwards, they were adjusted for the monthly temperature and precipitation data that were obtained with the climatic reconstruction and future projections. We did not carry out a sensitivity analysis as we think the data sequence is long enough to encompass inter- and intra-yearly variability.

**>> Line 374**: Terskol time period is already specified in Table 2 and line 370.

The text *"(1977−2013 with a data gap between 1990−1997)"* was deleted.

**>> Line 378-388**: This paragraph (and also L 396-400) with the listing of different dates and periods is a bit cumbersome to read. Maybe it would be better to indicate these periods in the anyway mentioned Fig. 10 and be more concise in the text.

The subsection was shortened to:

*"Especially during the last few decades, an accelerated warming trend has occurred, as the latest 10-year climatic interval exhibits a mean annual temperature anomaly of +0.5°C compared to the 1981−2010 mean. This makes it the warmest period in the reconstructed time series. For temperature, a clear sequence of colder and warmer intervals can be seen. Changes in precipitation show a sequence of drier and wetter periods (Fig. 8)."*

**>> Line 393-395 (and 494-495)**: The mass-balance and debris cover models were calibrated for the period 1967-2007 with the use of multiple tuning parameters to fit the observed surface mass-balance. The fact that no further dynamic calibration via mass-balance perturbations was necessary for this period cannot lead to conclusions about the model performance and accuracy.

See general comment 1.

**>> Future glacier evolution**: Like other reviewers, I do not understand how the GCM climate is used in this study. Why is the linear change necessary, why not applying the GCMs delta T and delta P directly?

We have now recreated the future climate forcing directly from available CMIP5 models for the grid cell closest to Djankuat Glacier. We therefore used a multi-model mean approach using 21 Global Circulation Models. We also applied a de-biasing procedure to match the future climate

forcing with the past, both concerning the trend and the variability. See general comment 1 from RC 1.